# Quantitative Analysis of the Effect of Density Ratio Estimation in Covariate Shift Adaptation

**Chenglin Yu**                                                     *clyycl@outlook.com*
*School of Computer Science*
*Wuhan University*

**Zhengyu Zhou**                                          *zzysince1999@gmail.com*
*School of Computer Science*
*Wuhan University*

**Weiwei Liu**[*]                                          *liuweiwei863@gmail.com*
*School of Computer Science*
*Wuhan University*

**Reviewed on OpenReview:** *https://openreview.net/forum?id=TtWsnXTYUV*

## Abstract

In supervised learning, it is essential to assume that the test sample and the training sample come from the same distribution. But in reality, this assumption is frequently broken, which can lead to subpar performance from the learned model. We examine the learning problem under *covariate shift*, in which the conditional distribution of labels given covariates does not change despite the covariate distribution shifting. Two-step procedures, which first compute the density ratio and then carry out importance-weighted empirical risk minimization, are a popular family of methods for addressing covariate shift. However, the two-step techniques' performance could degrade due to estimation error of the density ratio. Unfortunately, the extent of the density ratio estimation error that affects the accuracy of learning algorithms is rarely analyzed. This paper accordingly provides a quantitative answer to this question. Specifically, we formulate the two-step covariate adaptation methods as a meta-algorithm. We show that the effect of the density ratio estimation error on the excess risk bound of the meta algorithm is of the fourth order, i.e., $\mathcal{O}\left(\epsilon_1\left(\mathcal{G}, S_{s1}, S_t, \delta/2\right)^4\right)$, if the true risk satisfies a requirement known as the *derivative vanishing* property, where $\epsilon_1\left(\mathcal{G}, S_{s1}, S_t, \delta/2\right)$ is the convergence rate of the density ratio estimation algorithm, $\mathcal{G}$ is the density ratio function class, $S_{s1}$ and $S_t$ are the samples generated by training distribution and test distribution respectively, and $\delta/2$ is the confidence parameter. Moreover, we analyze the impact of two specific density ratio estimation algorithms, Kullback-Leibler Importance Estimation Procedure and Kernel unconstrained Least-Squares Importance Fitting, on the final classifier's generalization error. We also report the experimental results of two-step covariate shift adaptation with a toy classification dataset using KLIEP.

## 1 Introduction

A fundamental assumption in machine learning is that training and test sample are *independent and identical distributed (i.i.d.)*. Because it makes it simple to extrapolate conclusions from training data to test data, this assumption is essential to learning. It is possible to provide statistical guarantees on the performance of traditional machine learning algorithms, so long as the data can be modeled under this assumption (Christopher M & Nasser M, 2006; Shalev-Shwartz & Ben-David, 2014). Real-world data,

---

[*]Corresponding author

however, rarely meet this assumption(Zhou & Liu, 2023; Wang et al., 2024; Zhou & Liu, 2026). Patterns for engineering controls fluctuate due to the non-stationarity of environments (Raza et al., 2015; Shi & Liu, 2024; 2025); incoming signals used for natural language and image processing, bioinformatics, or econometric analyses change in distribution over time and seasonality (Liu et al., 2018); patients used for prognostic factor identification in clinical trials may not come from the target population due to sample selection bias (He et al., 2021).

*Covariate shift* is more common and realistic for real data when compared to the i.i.d. assumption. It states that the conditional distribution of the labels given features is the same for both the training and test data (Maldonado et al., 2023). However, the marginal distributions of the training and test data are different. In the case of cancer diagnosis, for instance, the training set might contain an excessive number of sick patients, frequently of a particular subtype that is endemic in the area where the data was collected. Similarly, the distribution of covariates in advertising systems is non-stationary because of temporal variations in user interest. Additionally, the covariate shift is feasible in numerous real-world applications, including speaker identification (Yamada et al., 2010), brain-computer interfacing (Li et al., 2010), emotion recognition (Jirayucharoensak et al., 2014), human activity recognition (Hachiya et al., 2012), and spam filtering (Bickel & Scheffer, 2007).

The model that is trained using standard machine learning techniques, like *empirical risk minimization (ERM)* (Vapnik, 1998; Schölkopf & Smola, 2002), is biased because of the difference between the training and test distributions under covariate shift. Many scholars suggest two-step methods to overcome this drawback: first, they estimate the ratio of test and training input (instance) densities, and then they minimize a weighted empirical loss (Sugiyama et al., 2011). In each case, the density ratio for a training example indicates the training example's importance to the test distribution. With such a mechanism, the examples which are important to the test distribution will be given large weights, and minimizing the weighted loss will inevitably result in a hypothesis that fits these examples well. Density ratio estimation is challenging, though, and as a result, estimation error may worsen learning outcomes in real-world scenarios (Yamada et al., 2011). Regretfully, it is still unclear how the estimation affects the final hypothesis's accuracy.

In order to comprehend the impact of density ratio estimation error on the ultimate hypothesis's performance, we pose the following query: *how much will the density ratio estimation error impact the accuracy of the hypothesis returned by the learning algorithm?* Inspired by the orthogonal statistical learning framework (Dylan J & Vasilis, 2023), we formulate the two-step approaches as a meta algorithm to address this question. We demonstrate that the impact of the density ratio estimation error on the excess risk bound of the meta algorithm is of the fourth order, i.e., $\mathcal{O}\left(\epsilon_1\left(\mathcal{G}, S_{s1}, S_t, \delta/2\right)^4\right)$, provided that the true risk satisfies a condition known as *derivative vanishing*, where $\epsilon_1\left(\mathcal{G}, S_{s1}, S_t, \delta/2\right)$ is the convergence rate of the density ratio estimation algorithm; $\mathcal{G}$ denotes the density ratio function class; $S_{s1}$ and $S_t$ represents samples coming from training and test distributions respectively; finally, $\delta/2$ is the confidence parameter. Moreover, based on our theoretical results, we investigate two commonly used density ratio estimation techniques: *Kullback-Leibler Importance Estimation Procedure (KLIEP)* (Sugiyama et al., 2008) and *Kernel unconstrained Least-Squares Importance Fitting (KuLSIF)* (Kanamori et al., 2012). Using *Importance Weighted Empirical Risk Minimization (IWERM)* for a linear hypothesis class, we derive excess risk bounds of the hypothesis for each of these estimation techniques. We also report the experimental results of two-step covariate shift adaptation with a toy classification dataset using KLIEP.

Our contributions are as folows: (1) a unified meta-algorithmic framework that formulates diverse two-step covariate shift adaptation methods as a single meta-algorithm (Algorithm 1), cleanly decomposing the excess risk into components arising from density ratio estimation and from learning on the weighted sample; (2) a precise quantification of error propagation under the noted assumptions (satisfied by square loss); (3) explicit excess risk bounds for classical estimators, where, while convergence rates for KLIEP and KuLSIF are known individually, integrating them into our framework yields a novel, comparative perspective by showing that their rates, when used in this two-step pipeline, fall short of the oracle $\mathcal{O}(n^{-1})$ rate.

## 2 Preliminaries

This section is organized as follows. First, in Section 2.1, we establish our notation, which is mostly standard. Then we recall covariate shift assumption and covariate shift adaptation in Section 2.2. Lastly, we present the two-step covariate shift adaptation approaches and formulate them as a meta algorithm in Section 2.3.

### 2.1 Notation

Let $\mathcal{Z} = \mathcal{X} \times \mathcal{Y}$ denote the sample space with $\mathcal{X} \subset \mathcal{R}^d$ ($d \in \mathcal{N}$) being the instance space and $\mathcal{Y} = \{0,1\}$ being the label space. For a function $f : \mathcal{X} \to \mathcal{R}$ and a distribution $\mathcal{D}$ over $\mathcal{X}$, the functional $L_p$-norm is defined as: $\|f\|_{L_p(\mathcal{D})} = (\mathbb{E}_{x \sim \mathcal{D}}|f(x)|^p)^{\frac{1}{p}}$. We define the *star hull* of $x \in \mathcal{X}$ as $\mathrm{star}\,(\mathcal{X}, x) = \{t \cdot x + (1-t) \cdot x' \mid x' \in \mathcal{X}, t \in [0,1]\}$. In addition, denote the convex hull of $\mathcal{X}$ as $\mathrm{conv}\,(\mathcal{X})$. We denote the transpose of the vector/matrix by the superscript $^\top$.

### 2.2 Covariate Shift Adaptation

Let $S_s = \{(x_i^s, y_i^s)\}_{i=1}^{n_s}$ be the i.i.d. training sample drawn from a training distribution $\mathcal{D}_s$ with density $p_s(x,y)$, which can be decomposed into the marginal distribution and the conditional probability distribution, i.e., $p_s(x,y) = p_s(x)p_s(y|x)$. Let $\{(x_i^t, y_i^t)\}_{i=1}^{n_t}$ be a test sample drawn from a test distribution $\mathcal{D}_t$ with density $p_t(x,y) = p_t(x)p_t(y|x)$. The corresponding marginal distributions are denoted as $\mathcal{D}_s^x$ and $\mathcal{D}_t^x$. Under the *covariate shift* assumption (Shimodaira, 2000), the input marginal distributions vary between training and test, i.e., $p_s(x) \neq p_t(x)$, but the conditional distribution of the labels (given features) exhibits invariance across training and test stages, i.e., $p_s(y|x) = p_t(y|x)$. The objective of covariate shift adaptation is to select a classifier $c : \mathcal{X} \to \{0,1\}$ out of a hypothesis class with a small risk with respect to the test distribution $\mathcal{D}_t$, $L_{\mathcal{D}_t}^{0-1}(h) := \mathbb{E}_{z \sim \mathcal{D}_t}[\mathbb{I}_{[c(x) \neq y]}]$, where $\mathbb{I}_{[\text{Boolean expression}]}$ is the indicator function (equal to 1 if the expression is true and 0 otherwise).

### 2.3 Two-Step Covariate Shift Adaptation Approaches

Let $\mathcal{H}$ be a class of functions from $\mathcal{X}$ to $[0,1]$ and let the risk $L_{\mathcal{D}_t}(h) := \mathbb{E}_{z \sim \mathcal{D}_t}[\ell(h(x), y)]$, where $\ell : [0,1] \times \mathcal{Y} \to \mathcal{R}^+$ denotes the loss function that measures the discrepancy between the predicted value $h(x)$ and the true output value $y$. In covariate shift adaptation, the only labeled sample we observe is $S_s = \{(x_i^s, y_i^s)\}_{i=1}^{n_s}$, which is drawn *i.i.d.* from $\mathcal{D}_s$. However, we need to find a minimizer $h^\star$ of risk with respect to $\mathcal{D}_t$, for which we only have *i.i.d.* draws from $\mathcal{D}_t^x$. Reweighing data to ensure that the reweighted distribution matches the test distribution is a popular method for dealing with covariate shift. In particular, if $g_0(x) = \frac{p_t(x)}{p_s(x)}$ is known, importance weighting can be used for the problem:

$$
\begin{aligned}
L_{\mathcal{D}_t}(h) &= \mathbb{E}_{z \sim \mathcal{D}_t}[\ell(h(x), y)] = \int_{\mathcal{X}} \ell(h(x), y)\, p_t(x)p_t(y|x)dx \\
&= \int_{\mathcal{X}} \ell(h(x), y)\, p_t(x)p_s(y|x)dx \\
&= \int_{\mathcal{X}} \frac{p_t(x)}{p_s(x)} \ell(h(x), y)\, p_s(x)p_s(y|x)dx = \mathbb{E}_{z \sim \mathcal{D}_s}[\ell(h(x), y) \cdot g_0(x)]
\end{aligned}
\tag{1}
$$

Let $\tilde{\ell}(h(x), g(x); z) := \ell(h(x), y) \cdot g(x)$, where $g : \mathcal{X} \to \mathcal{R}^+$ is a density ratio function. Denote $\mathbb{E}_{z \sim \mathcal{D}_s}[\tilde{\ell}(h(x), g(x); z)]$ by $\tilde{L}_{\mathcal{D}_s}(h, g)$. We then have $L_{\mathcal{D}_t}(h) = \tilde{L}_{\mathcal{D}_s}(h, g_0)$. The importance weighted empirical risk with respect to the training distribution is therefore an unbiased estimator of the risk with respect to the test distribution for any fixed $h \in \mathcal{H}$. The ability to precisely estimate the density ratio turns into a crucial component in the covariate shift adaptation process. Consequently, the majority of the research is focused on estimating the density ratio (Sugiyama et al., 2007b; Kanamori et al., 2009; Huang et al., 2006). Next, a predictive model in the training phrase is trained using the estimated density ratio, frequently with the help of the IWERM technique (Shimodaira, 2000; Sugiyama & Kawanabe, 2012). In order to analyze the effect of the density ratio estimator error on the final hypothesis returned by IWERM at a high level, we formulate the two-step covariate shift adaptation approaches as a meta algorithm presented as Algorithm 1.

---

**Algorithm 1** Two-step Covariate Shift Adaptation Meta Algorithm

---

**Input:** Sample set $S_s = \{(x_i^s, y_i^s)\}_{i=1}^{n_s} \sim \mathcal{D}_s$, $S_t = \{x_i^t\}_{i=1}^{n_t} \sim \mathcal{D}_t^x$

**Output:** $\hat{h}$, the prediction hypothesis.

1: Split $S_s$ into two subsets $S_{s1} = (x_i^s, y_i^s)_{i=1}^{n_{s1}}$, and $S_{s2} = S_s \backslash S_{s1}$.

2: Let $\hat{g}$ be the output of DRE($\mathcal{G}, S_{s1}, S_t$).

3: **return** $\hat{h}$, the output of IWERM($\mathcal{H}, S_{s2}; \hat{g}$).

---

In the two-step covariate shift adaptation meta algorithm, we first use a *Density Ratio Estimation (DRE)* algorithm to estimate the density ratio function $g_0$, and then plug the estimated density ratio into IWERM. For subsequent analysis, we give accurate definitions of DRE and IWERM with their convergence properties being assumed.

**Definition 1** (DRE algorithm and its convergence rate). *Let $\mathcal{G}$ be a density ratio function class from $\mathcal{X}$ to $\mathcal{R}^+$. Define $\|g\|_{\mathcal{G}} = \|g\|_{L_4(\mathcal{D}_s^x)}$. Given $S_{s1} \sim \mathcal{D}_s$, $S_t \sim \mathcal{D}_t^x$, let the algorithm DRE($\mathcal{G}, S_{s1}, S_t$) can output a density ratio function $\hat{g}$, for which*

$$\|\hat{g} - g_0\|_{\mathcal{G}} \le \epsilon_1 (\mathcal{G}, S_{s1}, S_t, \delta) \tag{2}$$

*with probability at least $1 - \delta$.*

**Definition 2** (IWERM algorithm and its convergence rate). *Suppose $h^\star$ is a minimizer of $\tilde{L}_{\mathcal{D}_s}(h, g_0)$. Given $S_{s2} \sim \mathcal{D}_s$ and $g \in \mathcal{G}$, let the algorithm IWERM($\mathcal{H}, S_{s2}; g$) can output a predictor $\hat{h} \in \mathcal{H}$, for which*

$$\tilde{L}_{\mathcal{D}_s}(\hat{h}, g) - \tilde{L}_{\mathcal{D}_s}(h^\star, g) \le \epsilon_2 (\mathcal{H}, S_{s2}, \delta; g) \tag{3}$$

*with probability at least $1 - \delta$.*

## 3 Main Results

We aim to quantitatively analyze the effect of density ratio estimation error ($\epsilon_1 (\mathcal{G}, S_{s1}, S_t, \delta)$) on the final hypothesis's excess risk, i.e., $\tilde{L}_{\mathcal{D}_s}(\hat{h}, g_0) - \tilde{L}_{\mathcal{D}_s}(h^\star, g_0)$. For convenience of illustration, we first give the definition of *directional derivative* on the vector space of functions.

**Definition 3** (Directional Derivative). *Let $\mathcal{T}$ be a vector space of functions. For a functional $T : \mathcal{T} \to \mathcal{R}$, the directional derivative of $T$ at a function $t \in \mathcal{T}$ in the direction $t' \in \mathcal{T}$ is defined as*

$$\partial_t T(t)[t'] = \frac{d}{d\mu} T(t + \mu t') \Big|_{\mu=0} = \lim_{\delta \to 0} \frac{T(t + \delta t') - T(t)}{\delta}.$$

*The $k$-th order directional derivative of $T$ at a function $t \in \mathcal{T}$ in the directions $t_1', \dots, t_k' \in \mathcal{T}$ is defined as*

$$\partial_t^k T(t)[t_1', \dots, t_k'] = \frac{\partial^k}{\partial \mu_1 \cdots \partial \mu_k} T(t + \mu_1 t_1' + \cdots + \mu_k t_k') \Big|_{\mu_1=0,\dots,\mu_k=0}.$$

*For a functional $T(h, g) : \mathcal{T} \times \mathcal{T} \to \mathcal{R}$, the partial derivative of $T$ with respect to $h$ in the direction $t_1'$ is*

$$\partial_h T(h, g)[t_1'] = \frac{d}{d\mu} T(h + \mu t_1', g) \Big|_{\mu=0} = \lim_{\delta \to 0} \frac{T(h + \delta t_1', g) - T(h, g)}{\delta}.$$

*The partial derivative of $T$ with respect to $g$ in the direction $t_2'$ is*

$$\partial_g T(h, g)[t_2'] = \frac{d}{d\mu} T(h, g + \mu t_2') \Big|_{\mu=0} = \lim_{\delta \to 0} \frac{T(h, g + \delta t_2') - T(h, g)}{\delta}.$$

*The second partial derivative of $T$ with respect to $h$ in the direction $t_1'$ then $h$ in the direction $t_2'$ is $\partial_h(\partial_h T(h, g)[t_1'])[t_2']$, denoted as $\partial_h^2 T(h, g)[t_1', t_2']$. The second partial derivative of $T$ with respect to $h$ in the direction $t_1'$ then $g$ in the direction $t_2'$ is $\partial_g(\partial_h T(h, g)[t_1'])[t_2']$, denoted as $\partial_g \partial_h T(h, g)[t_1', t_2']$. Similar definitions hold for $\partial_g^2 T(h, g)[t_1', t_2']$ and $\partial_h \partial_g T(h, g)[t_1', t_2']$.*

**Remark 1.** *The directional derivative defined in Definition 3 (also called the Gateaux derivative ) is a generalization of the directional derivative in multivariable calculus (Daryoush & Encyeh Dehghan, 2008).*

The first assumption we make is that the risk function has the so-called *derivative vanishing* property, which implies that some partial derivatives vanish.

**Assumption 1** (Derivative Vanishing Loss). *$\tilde{L}_{\mathcal{D}_s}(h, g)$ is derivative vanishing with respect to $\mathcal{H}$ and $\mathcal{G}$, that is,*

$$\partial_g \partial_h \tilde{L}_{\mathcal{D}_s}(h^\star, g_0)[h_1 - h^\star, g_1 - g_0] = 0 \quad \forall h_1 \in \mathcal{H}, \forall g_1 \in \mathcal{G}. \tag{4}$$

Our main theorem requires a first-order condition for the hypothesis class in the sense that $h^\star$ is a local minimizer, in addition to the derivative vanishing property.

**Assumption 2** (First Order Condition). *The minimizer for the risk satisfies the first-order condition:*

$$\partial_h \tilde{L}_{\mathcal{D}_s}(h^\star, g_0)[h_1 - h^\star] \geq 0 \quad \forall h_1 \in \text{star}(\mathcal{H}, h^\star). \tag{5}$$

Further, we need a few technical assumptions.

**Assumption 3.** *Define the norm over $\mathcal{H}$ as: $\|h\|_{\mathcal{H}} = \left(\mathbb{E}_{x \sim \mathcal{D}_s^x}[(2g_0(x)h(x))^2]\right)^{\frac{1}{2}}$. There exist constants $k_1, k_2$ such that the following holds:*

    a *The risk $\tilde{L}_{\mathcal{D}_s}$ is strongly convex with respect to the hypothesis: $\forall h_1 \in \mathcal{H}$ and $g_1 \in \mathcal{G}$,*

$$\partial_h^2 \tilde{L}_{\mathcal{D}_s}(\bar{h}, g)[h_1 - h^\star, h_1 - h^\star] \geq \lambda \|h_1 - h^\star\|_{\mathcal{H}}^{2k_1} - \kappa \|g_1 - g_0\|_{\mathcal{G}}^{2k_2} \quad \forall \bar{h} \in \text{star}(\mathcal{H}, h^\star). \tag{6}$$

    b *There exists a constant $\beta_1$, such that the following bound holds: $\forall h_1 \in \mathcal{H}$ and $\bar{h} \in \text{star}(\mathcal{H}, h^\star)$:*

$$\partial_h^2 \tilde{L}_{\mathcal{D}_s}(\bar{h}, g_0)[h_1 - h^\star, h_1 - h^\star] \leq \beta_1 \cdot \|h_1 - h^\star\|_{\mathcal{H}}^{2k_1}. \tag{7}$$

    c *There exists a constant $\beta_2$, such that the following bound holds: $\forall h_1 \in \text{star}(\mathcal{H}, h^\star)$, $g_1 \in \mathcal{G}$, and $\bar{g} \in \text{star}(\mathcal{G}, g_0)$:*

$$\left|\partial_g^2 \partial_h \tilde{L}_{\mathcal{D}_s}(h^\star, \bar{g})[h_1 - h^\star, g_1 - g_0, g_1 - g_0]\right| \leq \beta_2 \cdot \|h_1 - h^\star\|_{\mathcal{H}}^{k_1} \cdot \|g_1 - g_0\|_{\mathcal{G}}^{k_2}. \tag{8}$$

An observation is that for $k_1 = 1$ and $k_2 = 2$, Items b and c in Assumption 3 represent second-order and higher-order smoothness of the true risk, respectively. Moreover, Assumptions 1 to 3 are easily met whenever the square loss is applied to the hypothesis class prediction in order to determine the true risk. We shall discuss this in more depth in Section 3.1.

The main theorem is presented below.

**Theorem 1.** *Suppose $\mathbb{E}_{z \sim \mathcal{D}_s}[\tilde{\ell}(h(x), g(x); z)] = \tilde{L}_{\mathcal{D}_s}(h, g)$ satisfies Assumptions 1 to 3. Let $\hat{g}$ be the output of DRE($\mathcal{G}, S_{s1}, S_t$). We then have:*

*If $\beta_2$ in Item c of Assumption 3 is $0$, then the Algorithm 1 will produce a predictor $\hat{h}$ such that, with probability at least $1 - \delta$,*

$$\tilde{L}_{\mathcal{D}_s}(\hat{h}, g_0) - \tilde{L}_{\mathcal{D}_s}(h^\star, g_0) \leq \frac{\beta_1 \kappa}{2\lambda} \epsilon_1\left(\mathcal{G}, S_{s1}, S_t, \frac{\delta}{2}\right)^{2k_2} + \frac{\beta_1}{\lambda} \epsilon_2\left(\mathcal{H}, S_{s2}, \frac{\delta}{2}; \hat{g}\right) \tag{9}$$

*If $\beta_2$ in Item c of Assumption 3 is $> 0$, then the Algorithm 1 will produce a predictor $\hat{h}$ such that, with probability at least $1 - \delta$,*

$$\tilde{L}_{\mathcal{D}_s}(\hat{h}, g_0) - \tilde{L}_{\mathcal{D}_s}(h^\star, g_0) \leq \left(\frac{\beta_1 \kappa}{\lambda} + \frac{\beta_1 \beta_2^2}{2\lambda^2}\right) \epsilon_1\left(\mathcal{G}, S_{s1}, S_t, \frac{\delta}{2}\right)^{2k_2} + \frac{2\beta_1}{\lambda} \epsilon_2\left(\mathcal{H}, S_{s2}, \frac{\delta}{2}; \hat{g}\right) \tag{10}$$

*Proof of Theorem 1.* The proof is provided in Appendix B. □

**Remark 2.** *Theorem 1 shows that, for Algorithm 1, the impact of the unknown density ratio function on the prediction has $2k_2$-order growth: $\mathcal{O}\left(\epsilon_1\left(\mathcal{G}, S_{s1}, S_t, \frac{\delta}{2}\right)^{2k_2}\right)$. This suggests that if the desired oracle rate of Algorithm 1 with the true density ratio function being known is of order $\mathcal{O}(n^{-1})$, it suffices to take $\epsilon_1\left(\mathcal{G}, S_{s1}, S_t, \frac{\delta}{2}\right) = \mathcal{O}(n^{-\frac{1}{2k_2}})$. In addition, we can also analyze the excess risk for specific two-step domain adaptation algorithms using this theorem.*

### 3.1 Square Loss Satisfies Assumptions 1 to 3

We now show that the sufficient conditions introduced in Theorem 1 (Assumptions 1 to 3) are not merely abstract technicalities: they are naturally and exactly satisfied by the square loss under mild boundedness conditions. This concrete instantiation makes the error propagation result in Theorem 1 directly applicable to a core class of regression problems.

In particular, several basic properties of the weighted square loss, that is, $\tilde{\ell}(\zeta, \gamma; z) = (\zeta - y)^2 \cdot \gamma$ are obtained assuming $\exists h^\star \in \mathcal{H}$ such that $h^\star = \mathbb{E}[y|x]$. Then, we show that the true risk satisfies Assumptions 1 to 3 under a little more conditions posed on the hypothesis class and density ratio function class. Finally, a corollary with concrete constants is presented.

**Lemma 1.** *Suppose $\mathbb{E}[y|x] = h^\star(x)$. Then $\tilde{\ell}(\zeta, \gamma; z) = (\zeta - y)^2 \cdot \gamma$ has the following properties:*

1. *$\mathbb{E}[\nabla_\zeta \nabla_\gamma \tilde{\ell}(h^\star(x), g_0(x); z)|x] = 0$*

2. *$\mathbb{E}[\nabla_\zeta \tilde{\ell}(h^\star(x), g_0(x); z)(h(x) - h^\star(x))] = 0$
   $\forall h \in \mathrm{star}(\mathcal{H}, h^\star)$*

3. *$\mathbb{E}[\nabla^2_{\gamma\gamma} \nabla_\zeta \ell(h^\star(x), \bar{g}(x); z)|x] = 0 \quad \forall g \in \mathrm{star}(\mathcal{G}, g_0)$*

*Proof of Lemma 1.* The proof is provided in Appendix C. □

Properties in Lemma 1 and some additional conditions lead to Theorem 2:

**Theorem 2.** *Suppose $\mathbb{E}[y|x] = h^\star(x)$, $\sup_{x \in \mathcal{X}, h \in \mathcal{H}}|h(x)| \leq M_h$, $\forall g \in \mathcal{G}, x \in \mathcal{X}, g(x) \geq \eta > 0$, $g_0(x) \geq \eta$. Then $\tilde{L}_{\mathcal{D}_s}(h, g) = \mathbb{E}[\tilde{\ell}(h, g; z)]$ satisfies Assumptions 1 to 3, with $\lambda = \frac{1}{4}$, $k_1 = 1$, $k_2 = 2$, $\kappa = \frac{M_h^2}{2\eta^3}$, $\beta_1 = 1$ and $\beta_2 = 0$.*

*Proof of Theorem 2.* The proof is provided in Appendix D. □

Combining Theorem 1 and Theorem 2, we obtain the following corollary:

**Corollary 1.** *Suppose $\mathbb{E}[y|x] = h^\star(x)$, $\sup_{x \in \mathcal{X}, h \in \mathcal{H}}|h(x)| \leq M_h$, $\forall g \in \mathcal{G}, x \in \mathcal{X}, g(x) \geq \eta > 0$, $g_0(x) \geq \eta$. Algorithm 1 will then produce a predictor $\hat{h}$ such that, with probability at least $1 - \delta$,*

$$\tilde{L}_{\mathcal{D}_s}(\hat{h}, g_0) - \tilde{L}_{\mathcal{D}_s}(h^\star, g_0) \leq \frac{M_h^2}{\eta^3} \epsilon_1\left(\mathcal{G}, S_{s1}, S_t, \frac{\delta}{2}\right)^4 + 4\epsilon_2\left(\mathcal{H}, S_{s2}, \frac{\delta}{2}; \hat{g}\right) \tag{11}$$

Corollary 1 shows that, for Algorithm 1, when using square loss, the impact of the unknown density ratio function on the prediction has fourth-order growth: $\mathcal{O}(\epsilon_1\left(\mathcal{G}, S_{s1}, S_t, \frac{\delta}{2}\right)^4)$. This suggests that if the desired oracle rate with the true density ratio function being known is of order $\mathcal{O}(n^{-1})$, it suffices to take $\epsilon_1\left(\mathcal{G}, S_{s1}, S_t, \frac{\delta}{2}\right) = \mathcal{O}(n^{-\frac{1}{4}})$. In addition, Corollary 1 instantiates the coefficients before the two convergence rates in Theorem 1. We will instantiate the bound further in Section 4.

**Remark 3.** *The theoretical analysis of Corollary 1 translates into clear, quantitative guidance for practitioners: the precision required in the density ratio estimation step is less stringent than the final target accuracy for the predictor. A DRE method converging at a rate of roughly $n^{-\frac{1}{4}}$ can be sufficient to support a final predictor converging at the desirable $n^{-1}$ rate.*

# 4 Effect of Density Ratio Estimation Error for Linear Hypothesis Class

Our main theorem in Section 3 is stated at a high level of generality. In this section, we will examine two concrete estimation techniques: KLIEP and KuLSIF. Particularly, we first analyze the estimation error of IWERM on a linear hypothesis class. Then, we obtain the convergence properties of KLIEP and KuLSIF. It's worth mentioning that our bounds have concrete probabilities which are not explicitly stated in the original papers of those two algorithms. Finally, we plug the convergence rates into the general bound in Corollary 1 respectively.

## 4.1 Excess Risk Bound of IWERM

This section analyzes the excess risk bound when the hypothesis class is the composition of the sigmoid function $\text{sig}(t) = \frac{1}{1+\exp(-t)}$ over the class of linear functions, i.e., $\mathcal{H} = \{x \mapsto \text{sig}(\langle w, x \rangle) : \|w\|_2 \leq 1\}$. In particular, we derive the relationship between the risks of the true minimizer and estimated risk minimizer with the density ratio function being $g$. For convenience, we denote $\mathcal{F} = \{f : (x, y) \mapsto h(x) - y | h \in \mathcal{H}\}$. $N_X(\Delta, \mathcal{F})$ denotes the $\Delta$-covering number of $\mathcal{F}$ with respect to the metric $\rho_X(f, f') = \|f - f'\|_\infty$. We have the following theorem:

**Theorem 3.** *Let $\mathcal{H} = \{x \mapsto \text{sig}(\langle w, x \rangle) : \|w\|_2 \leq 1\}$ and $\ell(h(x), y) = (h(x) - y)^2$. Suppose $\forall x \in \mathcal{X}, \|x\|_2 \leq B$ and $\forall g \in \mathcal{G}, \sup_{x \in X} g(x) \leq M$. When given $S_{s2} \sim \mathcal{D}_s$ and any $g \in \mathcal{G}$, IWERM($\mathcal{H}, S_{s2}; g$) outputs a predictor $\hat{h} \in \mathcal{H}$ for which, with probability at least $1 - \delta$,*

$$\tilde{L}_{\mathcal{D}_s}(\hat{h}, g) - \tilde{L}_{\mathcal{D}_s}(h^\star, g)$$

$$\leq 8M \inf_{\Delta \in [0, D], N_X(\Delta; \mathcal{F}) \geq 10} \left\{ 2\Delta + \frac{2B\sqrt{d \log\left(1 + \frac{B}{2\Delta}\right)}}{\sqrt{n_2}} \right\} + 2M\sqrt{\frac{2\log\left(\frac{1}{\delta}\right)}{n_2}} \tag{12}$$

*where $D = \sup_{w, w' \in \mathbb{B}(0,1)} \sup_{x \in \mathcal{X}} |\text{sig}(\langle w, x \rangle) - \text{sig}(\langle w', x \rangle)|$ and $n_2$ is the size of $S_{s2}$.*

*Proof of Theorem 3.* The proof is provided in Appendix E. ☐

## 4.2 KLIEP

KLIEP estimates the density ratio $g_0(x) = \frac{p_t(x)}{p_s(x)}$ and models $g_0(x)$ using the following linear model:

$$\hat{g}(x) = \sum_{\ell=1}^{b} \alpha_\ell \varphi_\ell(x) \tag{13}$$

where $\{\alpha_\ell\}_{\ell=1}^{b}$ are the parameters to be learned from $S_{s1}$ and $S_t$, and $\{\varphi_\ell(x)\}_{\ell=1}^{b}$ are basis functions such that $\varphi_\ell(x) \geq 0$, for $x \in \mathcal{X}$ and $\ell = 1, 2, \ldots, b$. The target input density $p_t(x)$ can be estimated by $\hat{p}_t(x) = \hat{g}(x) p_s(x)$. Parameters $\{\alpha_\ell\}_{\ell=1}^{b}$ can be optimized by minimizing the Kullback-Leibler divergence between $p_t(x)$ and $\hat{p}_t(x)$:

$$\text{KL}[p_t(x) \| \hat{p}_t(x)] = \int_{\mathcal{X}} p_t(x) \log \frac{p_t(x)}{\hat{g}(x) p_s(x)} \mathrm{d}x = \int_{\mathcal{X}} p_t(x) \log \frac{p_t(x)}{p_s(x)} \mathrm{d}x - \int_{\mathcal{X}} p_t(x) \log \hat{g}(x) \mathrm{d}x \tag{14}$$

Based on the empirical approximations, the optimization problem can be formulated as follows.

$$\underset{\{\alpha_\ell\}_{\ell=1}^{b}}{\text{maximize}} \left[ \sum_{j=1}^{n_t} \log\left( \sum_{\ell=1}^{b} \alpha_\ell \varphi_\ell(x_j^t) \right) \right]$$
$$\text{subject to } \sum_{i=1}^{n_{s1}} \sum_{\ell=1}^{b} \alpha_\ell \varphi_\ell(x_i^s) = n_{s1} \text{ and } \alpha_1, \alpha_2, \ldots, \alpha_b \geq 0 \tag{15}$$

There is a global solution available for this convex optimization problem. We shall analyze how fast the resulting estimated density ratio function converge to the real density ratio function. For simplicity, assume

$n_{s1} = n_t = n_1$. Additionally, the Hellinger distance between $g$ and $g'$ with respect to $p_s(x)$ is defined as follows:

$$h_{p_s}(g, g') := \left( \int_{\mathcal{X}} \left( \sqrt{g(x)} - \sqrt{g'(x)} \right)^2 p_s(x)\, dx \right)^{\frac{1}{2}}. \tag{16}$$

**Lemma 2.** Let $a_0^{n_1} := \frac{1}{n_1} \sum_{i=1}^{n_1} g_0(x_i^s)$, $\gamma_{n_1} := \max\{ -\frac{1}{n_1} \sum_{j=1}^{n_1} \log(\hat{g}(x_j^t)) + \frac{1}{n_1} \sum_{j=1}^{n_1} \log(a_0^{n_1} g_0(x_j^t)), 0 \}$. The KLIEP algorithm then returns $\hat{g}$ such that $h_{p_s}(g_0, \hat{g}) = \mathcal{O}\left( n_1^{-\frac{1}{2+\omega}} \log\left( \frac{4}{\delta} \right) + \sqrt{\gamma_{n_1}} \right)$ with probability at least $1 - \delta$, where $0 < \omega < 2$ is a constant.

The proof of Lemma 2 can be easily adapted from Theorem 1 of (Sugiyama et al., 2008).

Note that the bound in Lemma 2 is with respect to Hellinger distance. We shall bridge the gap between Hellinger distance and $L4$-functional norm.

**Lemma 3.** Suppose $\forall g \in \mathcal{G}, \sup_{x \in \mathcal{X}} g(x) \leq M$, $g_0(x) \leq \eta_1$. Then $\|\hat{g} - g_0\|_{L_4(\mathcal{D}_s^x)} \leq 2^{\frac{1}{4}} (\eta_1 + M)^{\frac{3}{4}} h_{p_s}(\hat{g}, g_0)^{\frac{1}{2}}$.

*Proof of Lemma 3.* The proof is provided in Appendix F. □

Combining Lemma 2 and Lemma 3, we obtain the bound of the distance between the empirical density ratio function returned by KLIEP and the real density ratio function, presented as Theorem 4.

**Theorem 4.** Suppose $\forall g \in \mathcal{G}$, $x \in \mathcal{X}$, $g(x) \leq M$, $g_0(x) \leq \eta_1$. Then $\|\hat{g} - g_0\|_{L_4(\mathcal{D}_s^x)} \leq 2^{\frac{1}{4}} (\eta_1 + M)^{\frac{3}{4}} \mathcal{O}\left( \left( n_1^{-\frac{1}{2+\omega}} \log\left( \frac{4}{\delta} \right) + \sqrt{\gamma_{n_1}} \right)^{\frac{1}{2}} \right)$ with probability at least $1 - \delta$, where $0 < \omega < 2$ is a constant.

The generalization error difference between the hypothesis returned by IWERM using the real density ratio function and the one using the estimated density ratio function is bound by the following corollary.

**Corollary 2.** Let $\mathcal{H} = \{ x \mapsto \mathrm{sig}(\langle w, x \rangle) : \|w\|_2 \leq 1 \}$. Suppose $\mathbb{E}[y|x] = h^\star(x)$. $\forall g \in \mathcal{G}, x \in \mathcal{X}, 0 < \eta \leq g(x) \leq M$, $\eta \leq g_0(x) < \eta_1$ and $\|x\|_2 \leq B, \forall x \in \mathcal{X}$. Algorithm 1 will then produce a predictor $\hat{h}$ such that, with probability at least $1 - \delta$,

$$\tilde{L}_{\mathcal{D}_s}(\hat{h}, g_0) - \tilde{L}_{\mathcal{D}_s}(h^\star, g_0) \leq \frac{1}{\eta^3} \left( 2(\eta_1 + M)^3 \mathcal{O}\left( \left( n_1^{-\frac{1}{2+\omega}} \log(\frac{8}{\delta}) + \sqrt{\gamma_{n_1}} \right)^2 \right) \right)$$

$$+ \left( 32M \inf_{\substack{\Delta \in [0, D] \\ N_X(\Delta; \mathcal{F}) \\ \geq 10}} \left\{ 2\Delta + \frac{2B\sqrt{d \log\left( 1 + \frac{B}{2\Delta} \right)}}{\sqrt{n_2}} \right\} + 8M\sqrt{\frac{2 \log\left( \frac{2}{\delta} \right)}{n_2}} \right) \tag{17}$$

where $D = \sup_{w, w' \in \mathbb{B}(0,1)} \sup_{x \in \mathcal{X}} |\mathrm{sig}(\langle w, x \rangle) - \mathrm{sig}(\langle w', x \rangle)|$, $0 < \omega < 2$ is a constant.

*Proof of Corollary 2.* The result follows from Corollary 1, Theorems 3 and 4. □

Corollary 2 shows that, the impact of KLIEP on Algorithm 1 is $\mathcal{O}\left( n_1^{-\frac{2}{2+\omega}} \right)$. Since $0 < \omega < 2$, we can tell that $-1 < -\frac{2}{2+\omega} < -\frac{1}{2}$. It implies that there is still a distance between the prediction error rate that can be achieved by KLIEP and fast rate, i.e., $\mathcal{O}(n^{-1})$.

**Remark 4.** *The speed with which a learning algorithm converges as it is presented with more data is a central problem in machine learning. Alexey Chervoonenkis and Vladimir Vapnik present quantitiave bounds on the deviation between the empirical and expected risk as a function of the sample size n (Vapnik, 1998). According to Vapnik (1998), in Vapnik's book co-authored by Chervonenkis VN & A Ya (1979), they present 'slow' and 'fast' bounds for Empirical Risk Minimization (ERM) when used with 0-1 loss. They show that in*

*the realizable or 'optimistic' case (where there is a predictor in the hypothesis class that almost surely predicts correctly, so that the minimum achievable risk is zero) one can achieve fast $\mathcal{O}(1/n)$ convergence, as opposed to the 'pessimistic' case where one does not have such a predictor in the hypothesis class and the best uniform bound is $\mathcal{O}(1/\sqrt{n})$ (typically so-called slow rate, page 127 in Vapnik (1998)). This difference is important because if one is in such a 'fast rate' regime, one can achieve good performance with less data.*

### 4.3 KuLSIF

Analogous to KLIEP, KuLSIF also models $g_0(x)$ using a linear model:

$$\hat{g}(x) = \sum_{\ell=1}^{b} \alpha_\ell \varphi_\ell(x) \tag{18}$$

where the parameters to be learned from $S_{s1}$ and $S_t$ are $\{\alpha_\ell\}_{\ell=1}^{b}$. The non-negativity condition $\varphi_\ell(x) \geq 0$ is satisfied by the basis functions $\varphi_\ell, \ell = 1, \ldots, b$.

KuLSIF assumes that the model for the density ratio, i.e., $\mathcal{G}$, is a Reproducing Kernel Hilbert Space (RKHS) endowed with a kernel function $k$ on $\mathcal{X} \times \mathcal{X}$, the inner product and the norm on $\mathcal{G}$ being denoted as $\langle \cdot, \cdot \rangle_{\mathcal{G}1}$ and $\|\cdot\|_{\mathcal{G}1}$, respectively[1]. The optimal solution to

$$\min_{g} \frac{1}{2n_{s1}} \sum_{i=1}^{n_{s1}} g\left(x_i^s\right)^2 - \frac{1}{n_t} \sum_{j=1}^{n_t} g\left(x_j^t\right) + \frac{\lambda}{2} \|g\|_{\mathcal{G}1}^2 \quad \text{s.t.} \quad g \in \mathcal{G}$$

yields the estimator $\hat{g}$. In order to prevent overfitting, the regularization parameter $\lambda(\geq 0)$ is introduced along with the regularization term $\frac{\lambda}{2}\|g\|_{\mathcal{G}1}^2$.

According to Theorem 1 of (Kanamori et al., 2012), the KuLSIF estimator is given as:

$$\hat{g}(x) = \sum_{i=1}^{n_{s1}} \bar{\alpha}_i k(x, x_i^s) + \frac{1}{n_t \lambda} \sum_{j=1}^{n_t} k(x, x_j^t) \tag{19}$$

The coefficients $\bar{\alpha} = (\bar{\alpha}_1, \ldots, \bar{\alpha}_{n_{s1}})^T$ are given by the solution of the linear equation:

$$\left(\frac{1}{n_{s1}} K_{11} + \lambda I_{n_{s1}}\right) \alpha = -\frac{1}{n_{s1} n_t \lambda} K_{12} \mathbf{1}_{n_t} \tag{20}$$

where $I_{n_{s1}}$ is the $n_{s1}$ by $n_{s1}$ identity matrix and $\mathbf{1}_{n_t}$ is the column vector defined as $\mathbf{1}_{n_t} = (1, \ldots, 1)^T \in \mathcal{R}^{n_t}$, $K_{11}$ and $K_{12}$ are the sub-matrices of the Gram matrix: $(K_{11})_{ii'} = k(x_i^s, x_{i'}^s)$, $(K_{12})_{ij} = k(x_i^s, x_j^t)$, where $i, i' = 1, \ldots, n_{s1}, j, j' = 1, \ldots, n_t$. We shall analyze how fast the resulting estimated density ratio function converges to the real density ratio function.

**Lemma 4.** *Assume $\mathcal{G}$ is a RKHS endowed with the kernel function $k$ defined on $\mathcal{X} \times \mathcal{X}$. The norm and inner product on $\mathcal{G}$ are denoted by $\|\cdot\|_{\mathcal{G}1}$ and $\langle \cdot, \cdot \rangle_{\mathcal{G}1}$. Suppose that $\sup_{x \in \mathcal{X}} k(x, x) < \infty$. Let $\mathcal{G}_M = \{g \in \mathcal{G} | \|g\|_{\mathcal{G}1} < M\}$, and that the bracketing entropy $H_B(t, \mathcal{G}_M, \mathcal{D}_s^x)$ is bounded above by $\mathcal{O}(\frac{M}{t})^\gamma$, where $\gamma$ is a constant satisfying $0 < \gamma < 2$, and $t$ is an arbitrary number such that $1 - \frac{2}{2+\gamma} < t < 1$. Set the regularization parameter $\lambda = \lambda_{n_{s1}, n_t}$ such that*

$$\lim_{n_{s1}, n_t \to \infty} \lambda_{n_{s1}, n_t} = 0, \lambda_{n_{s1}, n_t}^{-1} = \mathcal{O}\left(\left(n_{s1} \wedge n_t\right)^{1-t}\right) (n_{s1}, n_t \to \infty)$$

*where $n_{s1} \wedge n_t = \min\{n_{s1}, n_t\}$. Then, for $\frac{p_t}{p_s} = g_0 \in \mathcal{G}$, we have $\|\hat{g} - g_0\|_{L_2(\mathcal{D}_s^x)} = \mathcal{O}\left(c^2 \log(\frac{c}{\delta}) \lambda_{n_s, n_t}^{\frac{1}{2}}\right)$ with probability at least $1 - \delta$.*

The proof of Lemma 4 can be easily adapted from Theorem 2 of (Kanamori et al., 2012).

Note that the bound in Lemma 4 is not with respect to $L4$-functional norm either, but with respect to $L2$-functional norm. We bridge the gap between them.

---

[1]To avoid confusion with the previous notation $\|\cdot\|_{\mathcal{G}}$ defined in Definition 1

**Lemma 5.** *Suppose $\forall g \in \mathcal{G}, \sup_{x \in \mathcal{X}} g(x) \leq M$. Let $M_\delta = c^2 \log(\frac{c}{\delta})$, where $c$ is the same constant in Lemma 4. Then, $\|\hat{g} - g_0\|^4_{L_4(\mathcal{D}^x_s)} \leq (M + M_\delta)^2 \|\hat{g} - g_0\|^2_{L_2(\mathcal{D}^x_s)}$.*

*Proof of Lemma 5.* The proof is provided in Appendix G. □

Combining Lemma 4 and Lemma 5, we obtain the bound of the distance between the estimated density ratio function returned by KuLSIF and the real density ratio function, presented as Theorem 5.

**Theorem 5.** *Suppose $\forall g \in \mathcal{G}, \sup_{x \in \mathcal{X}} g(x) \leq M$. Then, $\|\hat{g} - g_0\|_{L_4(\mathcal{D}^x_s)} \leq \mathcal{O}\left(c^2 \log(\frac{c}{\delta})\lambda^{\frac{1}{4}}_{n_{s1},n_t}\right)$ with probability at least $1 - \delta$, where $c$ is a constant.*

*Proof of Theorem 5.* The proof is provided in Appendix H. □

The following corollary provides a bound on the difference between the generalization error of the hypothesis returned by IWERM using estimated density ratio function by KuLSIF and true risk minimizer in $\mathcal{H}$.

**Corollary 3.** *Let $\mathcal{H} = \{x \mapsto \mathrm{sig}(\langle w, x \rangle) : \|w\|_2 \leq 1\}$. Suppose $\mathbb{E}[y|x] = h^\star(x)$. $\forall g \in \mathcal{G}, x \in \mathcal{X}, 0 < \eta \leq g(x) \leq M$, $\eta \leq g_0(x) \leq M$ and $\|x\|_2 \leq B, \forall x \in \mathcal{X}$. Algorithm 1 will then produce a predictor $\hat{h}$ such that, with probability at least $1 - \delta$,*

$$\tilde{L}_{\mathcal{D}_s}(\hat{h}, g_0) - \tilde{L}_{\mathcal{D}_s}(h^\star, g_0) \leq \frac{1}{\eta^3} \mathcal{O}\left(\left(c^2 \log\frac{c}{\delta}\right)^4 \cdot \lambda_{n_{s1},n_t}\right) +$$

$$\left(32M \inf_{\substack{\Delta \in [0,D], \\ N_X(\Delta;\mathcal{F}) \\ \geq 10}} \left\{2\Delta + \frac{2B\sqrt{d\log\left(1 + \frac{B}{2\Delta}\right)}}{\sqrt{n_2}}\right\} + 8M\sqrt{\frac{2\log\left(\frac{2}{\delta}\right)}{n_2}}\right) \tag{21}$$

*where $D = \sup_{w,w' \in \mathbb{B}(0,1)} \sup_{x \in \mathcal{X}} |\mathrm{sig}(\langle w, x \rangle) - \mathrm{sig}(\langle w', x \rangle)|$, and $\lambda_{n_{s1},n_t}$ is the regularization parameter which satisfies the following condition:*

$$\lim_{n_{s1},n_t \to \infty} \lambda_{n_{s1},n_t} = 0, \lambda^{-1}_{n_{s1},n_t} = \mathcal{O}\left((n_{s1} \wedge n_t)^{1-t}\right) (n_{s1}, n_t \to \infty) \tag{22}$$

*where $t$ is an arbitrary number such that $1 - \frac{2}{2+\gamma} < t < 1$ ($\gamma$ is a constant satisfying $0 < \gamma < 2$).*

*Proof of Corollary 3.* The result follows from Corollary 1, Theorems 3 and 5. □

Corollary 3 shows that, the impact of KuLSIF on Algorithm 1 is $\mathcal{O}\left((n_{s1} \wedge n_t)^{t-1}\right)$. Since $1 - \frac{2}{2+\gamma} < t < 1$, we have $-1 < -\frac{2}{2+\gamma} < t - 1 < 0$. Therefore, two-step covariate shift adaptation for KuLSIF's convergence rate does not reach the fast rate, i.e., $\mathcal{O}(n^{-1})$.

## 5 Experiments

In this section, we compare the classification performance of *exponentially-flattened importance weighted ERM(EIWERM (Shimodaira, 2000))* using density ratio estimates returned by KLIEP with different kernel widths and true density ratios.

Let us consider a two-dimensional binary classification problem. The class posterior probabilities are defined by

$$p(y = +1|x) = \frac{\tanh(x^{(1)} + \min(0, x^{(2)})) + 1}{2}, \tag{23}$$

where $p(y = -1|x) = 1 - p(y = +1|x)$ and $x = (x^{(1)}, x^{(2)})^\top$. Let the training and test instance densities be

$$p_s(x) = \frac{1}{2}\Phi(x; \mathbf{m}_1, \Sigma_1) + \frac{1}{2}\Phi(x; \mathbf{m}_2, \Sigma_2),$$

$$p_t(x) = \frac{1}{2}\Phi(x; \mathbf{m}_3, \Sigma_3) + \frac{1}{2}\Phi(x; \mathbf{m}_4, \Sigma_4), \tag{24}$$

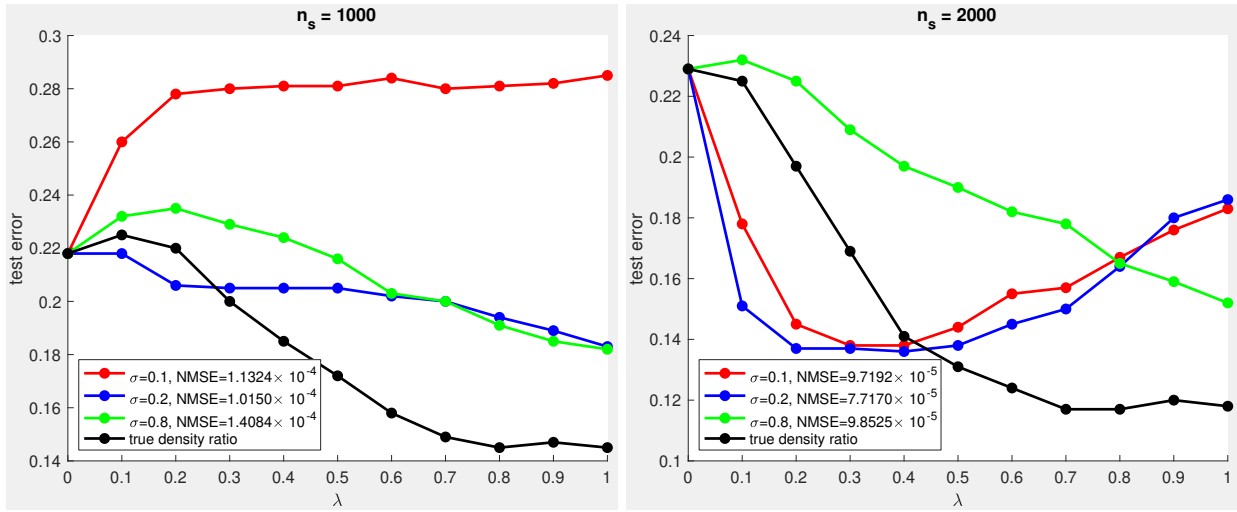

Figure 1: Test error (misclassification rate) as functions of the tuning parameter $\lambda$ on different density ratio estimates and real density ratio.

where $\Phi(x; \mathbf{m}, \Sigma)$ is the multivariate normal density with mean $\mathbf{m}$ and covariance matrix $\Sigma$, and $\mathbf{m}_1 = (-2, 3)^\top$, $\mathbf{m}_2 = (2, 3)^\top$, $\mathbf{m}_3 = (0, -1)^\top$, $\mathbf{m}_4 = (4, -1)^\top$, $\Sigma_1 = \Sigma_2 = \begin{pmatrix} 1 & 0 \\ 0 & 2 \end{pmatrix}$, $\Sigma_3 = \Sigma_4 = \begin{pmatrix} 1 & 0 \\ 0 & 1 \end{pmatrix}$. We create training instances $\{x_i^s\}_{i=1}^{n_s}$ following $p_s(x)$ and training labels $\{y_i^s\}_{i=1}^{n_s}$ following $p(y|x_i^s)$). Similarly, we create $n_t = 1000$ test instances $\{x_i^t\}_{i=1}^{n_t}$ following $p_t(x)$ and test labels $\{y_i^t\}_{i=1}^{n_t}$ following $p(y|x_i^t)$). This is a typical example of covariate shift.

**Setup.** For learning, we employ the following linear model:

$$\hat{f}(x; \theta) = \theta_0 + \sum_{i=1}^{d} \theta_i x^{(i)}, \tag{25}$$

where $d$ is the input dimension and $x^{(i)}$ is the $i$-th element of $x$. The parameter vector $\theta$ is learned by EIWERM:

$$\hat{\theta}_{EIWERM} \equiv \arg\min_\theta \left[ \frac{1}{n_s} \sum_{i=1}^{n_s} g(x_i^s)^\lambda \left( \hat{f}(x_i^s; \theta) - y_i^s \right)^2 \right], \tag{26}$$

where $0 \le \lambda \le 1$ is a hyperparameter controlling the trade-off between consistency and stability of EIWERM and $g$ is the density ratio function. The above minimizer $\hat{\theta}_{EIWERM}$ is analytically given by

$$\hat{\theta}_{EIWERM} = (X^\top D^\lambda X)^{-1} X^\top D^\lambda y, \tag{27}$$

where

$$X = \begin{pmatrix} 1 & x_1^{s\top} \\ 1 & x_2^{s\top} \\ \vdots & \vdots \\ 1 & x_{n_s}^{s\top} \end{pmatrix}, \tag{28}$$

$D$ is the diagonal matrix with the $i$-th diagonal element $D_{i,i} = g(x_i^s)$ and $y = (y_1^s, y_2^s, \ldots, y_{n_s}^s)^\top$. The classification result $\hat{y}$ of a test instance $x^t$ is obtained by:

$$\hat{y} = \text{sgn}(\hat{f}(x^t; \hat{\theta}_{EIWERM})), \tag{29}$$

where $\text{sgn}(\cdot)$ denotes the sign of a scalar. We use values $\lambda \in \{0, 0.1, 0.2, \cdots, 1.0\}$ to vary the trade-offs between consistency and stability of EIWERM.

The performance of KLIEP depends on the choice of the basis functions $\{\varphi_\ell(x)\}_{\ell=1}^b$ . Following Sugiyama et al. (2008), we use a Gaussian kernel model centered at a subset of $\{x_i^t\}_{i=1}^{n_t}$ for computational efficiency, i.e.,

$$\hat{g}(x) = \sum_{\ell=1}^b \alpha_\ell K_\sigma(x, c_\ell), \tag{30}$$

where $K_\sigma(x, x')$ is the Gaussian kernel with kernel width $\sigma$:

$$K_\sigma(x, x') = e^{-\frac{\|x - x'\|}{2\sigma^2}},$$

$c_\ell$ is a template instance randomly chosen from $\{x_i^t\}_{i=1}^{n_t}$ and $b(\leq n_t)$ is a prefixed number, as referenced on Eq. (7) of Sugiyama et al. (2008). We use values $\sigma \in \{0.1, 0.2, 0.8\}$ to vary the quality of the density ratio estimates produced by KLIEP, which is evaluated by the *normalized mean squared error(NMSE)*:

$$\text{NMSE} := \frac{1}{n_s} \sum_{i=1}^{n_s} \left( \frac{\hat{g}(x_i^s)}{\sum_{i'=1}^{n_s} \hat{g}(x_{i'}^s)} - \frac{g_0(x_i^s)}{\sum_{i'=1}^{n_s} g_0(x_{i'}^s)} \right)^2. \tag{31}$$

**Results.**  Fig. 1 depicts the test errors for $n_s = 1000$ and 2000. It is illustrated that the test error of the predictor returned by EIWERM for most of the $\lambda$ values decreases when NMSE decreases. The predictor obtained using real density ratio achieves lower test error compared with that obtained using density ratio estimates when $\lambda$ is greater than 0.5. The results show that reducing the density ratio estimation error is helpful for reducing the test error.

## 6  Related Work

### 6.1  Density Ratio Estimation

When there is a covariate shift, empirical risk minimization may not work as planned because the training and test distributions differ. In order to reduce the impact of covariate shift, importance weighting is utilized (Sugiyama et al., 2007a), which results in IWERM. The importance is frequently related to the ratio of the test and training instance densities. Thus, the accuracy of the density ratio estimation becomes critical to the covariate shift adaptation process.

Density ratio estimation algorithms are divided into two categories. The first involves estimating the two probability densities independently first, and then calculating the ratio of the estimated densities. However, it is well-known that density estimation is a challenging problem (Kanamori et al., 2010). As a result, this naive strategy might not work. Estimating the density ratio directly, without estimating the densities, is the second method. Various direct density-ratio estimation methods can be divided into four types: Moment Matching, Probabilistic Classification, Density Fitting and Density-Ratio Fitting (Masashi et al., 2012). Of all the direct methods, KLIEP (Sugiyama et al., 2008) and KuLSIF (Kanamori et al., 2012) are two commonly used ones which are computationally efficient and comparable to others in terms of performance (Kato & Teshima, 2021; Tiao et al., 2021). We include them as our analysis.

### 6.2  The Analysis of Effect of Density Ratio Estimation Error

The closest previous work to ours is Cortes et al. (2008). They consider the problem of figuring out how much the weight estimation error can impact the precision of the hypothesis that a weight-sensitive learning problem returns in the *missing at random (MAR)* bias problem. Their theoretical results differ from our work in three aspects.

First, MAR bias and covariate shift differ significantly from one another. Compared to covariate shift, the MAR bias places more stringent requirements on the relationship between the training and test distributions. In other words, all cases of MAR bias can be modeled as covariate shift, but some cases of covariate shift cannot result from MAR bias. Consider the data analysis problem with a brain-computer interface, where it

is known that as experiments progress — the subjects get tired, the sensor configuration varies, etc. — the distribution over incoming signals will alter. In this example, a covariate shift that cannot be represented by sample selection bias occurs. The relationship between covariate shift and MAR bias can be further referred to Appendix I.1 and I.3.

Second, the risks analyzed in Cortes et al. (2008) and this paper are different. Let $\hat{h}_{g_0}$ denote the predictor returned by IWERM($\mathcal{H}, S_{s2}; g_0$). Theorems 2 and 4 in Cortes et al. (2008) provide bounds on the difference between the generalization error of $\hat{h}_{g_0}$ and $\hat{h}$ (i.e., $\tilde{L}_{\mathcal{D}_s}(\hat{h}, g_0) - \tilde{L}_{\mathcal{D}_s}(\hat{h}_{g_0}, g_0)$). In contrast, we consider the difference between the generalization error of the hypothesis returned by IWERM based on the estimated density ratio and the best one in the hypothesis class (i.e. $\tilde{L}_{\mathcal{D}_s}(\hat{h}, g_0) - \tilde{L}_{\mathcal{D}_s}(h^\star, g_0)$). For the weight estimation error, they only consider the weights for the training sample, while our analysis consider the difference between the real density ratio function and the estimated density ratio function. Therefore, the analysis in Cortes et al. (2008) is orthogonal to our work.

Third, the techniques of deriving bounds are different. Cortes et al. (2008) derives bounds via the *distributional stability* of kernel-based regularization algorithms. In contrast, we derive excess risk bounds via the *derivative vanishing* property of the true risk. Details can be found in Appendix I.3.

Reddi et al. (2015) is another work that is relevant to ours. Using the unweighted solution as a variance-reducing proxy for the correct weighed solution, this paper proposes a doubly robust estimator to address the problem that many covariate shift correction algorithms face: the variance increases significantly whenever samples are reweighted. Moreover, they analyze the generalization bounds for the Penalized Risk Minimization (PRM) estimation algorithm based on a meta theorem. However, they only give the generalization bound of the final hypothesis. The effect of the density ratio estimation error on the learning algorithm is not explicitly illustrated by Reddi et al. (2015).

## 7 Conclusion

KLIEP and KuLSIF are two classical and fundamental density ratio estimation approaches in two-step covariate shift adaptation and still the effect of the estimation error on the final classifier's accuracy is comparatively less explored. To study this problem, we first formulate the two-step covariate shift adaptation approaches as a meta algorithm. We then show that if the true risk satisfies a condition called *derivative vanishing* property, the impact of the density ratio estimation error on the excess risk bound of the meta algorithm is of the fourth order. Furthermore, we analyze the convergence rates of KLIEP and KuLSIF with specific probability and give an upper bound of estimation error of IWERM for a linear hypothesis class. Finally, the excess risk bounds for the two density ratio estimation algorithms are obtained and the quantitative effects of the density ratio estimation error are illustrated. Numerical experiments collaborate the theoretical findings.

A natural question is whether the derivative-vanishing property holds for other common losses (e.g., logistic, hinge) or more complex model classes. While beyond the scope of this work—which focuses on establishing a clear foundation with a canonical loss—we view this as a valuable direction for future investigation.

### Acknowledgments

This work is supported by the Key R&D Program of Hubei Province under Grant 2024BAB038, the National Key R&D Program of China under Grant 2023YFC3604702.

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

## Appendix

In appendices, we provide proofs of Theorem 1, Lemma 1, Theorem 2, Theorem 3, Lemma 3, Lemma 5 and Theorem 5 in the main paper. For completeness, we present the necessary assumptions, lemmas and theorems in the main paper again. In addition, we provide related work in detail.

## A    Assumptions

**Assumption 1** (Derivative Vanishing Loss). *$\tilde{L}_{\mathcal{D}_s}(h, g)$ is derivative vanishing with respect to $\mathcal{H}$ and $\mathcal{G}$, that is,*

$$\partial_g \partial_h \tilde{L}_{\mathcal{D}_s}(h^\star, g_0)[h_1 - h^\star, g_1 - g_0] = 0 \quad \forall h_1 \in \mathcal{H}, \forall g_1 \in \mathcal{G}. \tag{32}$$

In addition to derivative vanishing property, our main theorem requires a first-order condition for the hypothesis class in the sense that $h^\star$ is a local minimizer.

**Assumption 2** (First Order Condition). *The minimizer for the risk satisfies the first-order condition:*

$$\partial_h \tilde{L}_{\mathcal{D}_s}(h^\star, g_0)[h_1 - h^\star] \geq 0 \quad \forall h_1 \in \text{star}(\mathcal{H}, h^\star). \tag{33}$$

Further, we need a few technical assumptions.

**Assumption 3.** *Define the norm over $\mathcal{H}$ as: $\|h\|_{\mathcal{H}} = \left(\mathbb{E}_{x \sim \mathcal{D}_s^x}[(2g_0(x)h(x))^2]\right)^{\frac{1}{2}}$. There exist constants $k_1, k_2$ such that the following holds:*

a  The risk $\tilde{L}_{\mathcal{D}_s}$ is strongly convex with respect to the hypothesis: $\forall h_1 \in \mathcal{H}$ and $g_1 \in \mathcal{G}$,

$$\partial_h^2 \tilde{L}_{\mathcal{D}_s}(\bar{h}, g)[h_1 - h^\star, h_1 - h^\star] \geq \lambda \|h_1 - h^\star\|_{\mathcal{H}}^{2k_1} - \kappa \|g_1 - g_0\|_{\mathcal{G}}^{2k_2} \quad \forall \bar{h} \in \text{star}\,(\mathcal{H}, h^\star)\,. \tag{34}$$

b  There exists a constant $\beta_1$, such that the following bound holds: $\forall h_1 \in \mathcal{H}$ and $\bar{h} \in \text{star}\,(\mathcal{H}, h^\star)$:

$$\partial_h^2 \tilde{L}_{\mathcal{D}_s}(\bar{h}, g_0)[h_1 - h^\star, h_1 - h^\star] \leq \beta_1 \cdot \|h_1 - h^\star\|_{\mathcal{H}}^{2k_1}. \tag{35}$$

c  There exists a constant $\beta_2$, such that the following bound holds: $\forall h_1 \in \text{star}\,(\mathcal{H}, h^\star)\,,\ g_1 \in \mathcal{G},$ and $\bar{g} \in \text{star}\,(\mathcal{G}, g_0)$:

$$\left| \partial_g^2 \partial_h \tilde{L}_{\mathcal{D}_s}(h^\star, \bar{g}) [h_1 - h^\star, g_1 - g_0, g_1 - g_0] \right| \leq \beta_2 \cdot \|h_1 - h^\star\|_{\mathcal{H}}^{k_1} \cdot \|g_1 - g_0\|_{\mathcal{G}}^{k_2}. \tag{36}$$

**Lemma 4.** *Assume $\mathcal{G}$ is a RKHS endowed with the kernel function $k$ defined on $\mathcal{X} \times \mathcal{X}$. The norm and inner product on $\mathcal{G}$ are denoted by $\|\cdot\|_{\mathcal{G}1}$ and $\langle \cdot, \cdot \rangle_{\mathcal{G}1}$. Suppose that $\sup_{x \in \mathcal{X}} k(x, x) < \infty$. Let $\mathcal{G}_M = \{g \in \mathcal{G} | \|g\|_{\mathcal{G}1} < M\}$, and that the bracketing entropy $H_B(t, \mathcal{G}_M, \mathcal{D}_s^x)$ is bounded above by $\mathcal{O}(\frac{M}{t})^\gamma$, where $\gamma$ is a constant satisfying $0 < \gamma < 2$, and $t$ is an arbitrary number such that $1 - \frac{2}{2+\gamma} < t < 1$. Set the regularization parameter $\lambda = \lambda_{n_{s1}, n_t}$ such that*

$$\lim_{n_{s1}, n_t \to \infty} \lambda_{n_{s1}, n_t} = 0, \lambda_{n_{s1}, n_t}^{-1} = \mathcal{O}\left( (n_{s1} \wedge n_t)^{1-t} \right) (n_{s1}, n_t \to \infty)$$

*where $n_{s1} \wedge n_t = \min\{n_{s1}, n_t\}$. Then, for $\frac{p_t}{p_s} = g_0 \in \mathcal{G}$, we have $\|\hat{g} - g_0\|_{L_2(\mathcal{D}_s^x)} = \mathcal{O}\left( c^2 \log(\frac{c}{\delta}) \lambda_{n_s, n_t}^{\frac{1}{2}} \right)$ with probability at least $1 - \delta$.*

**Definition 1** (DRE algorithm and its convergence rate). *Let $\mathcal{G}$ be a density ratio function class from $\mathcal{X}$ to $\mathcal{R}^+$. Define $\|g\|_{\mathcal{G}} = \|g\|_{L_4(\mathcal{D}_s^x)}$. Given $S_{s1} \sim \mathcal{D}_s$, $S_t \sim \mathcal{D}_t^x$, let the algorithm $DRE(\mathcal{G}, S_{s1}, S_t)$ can output a density ratio function $\hat{g}$, for which*

$$\|\hat{g} - g_0\|_{\mathcal{G}} \leq \epsilon_1 (\mathcal{G}, S_{s1}, S_t, \delta) \tag{37}$$

*with probability at least $1 - \delta$.*

**Definition 2** (IWERM algorithm and its convergence rate). *Suppose $h^\star$ is a minimizer of $\tilde{L}_{\mathcal{D}_s}(h, g_0)$. Given $S_{s2} \sim \mathcal{D}_s$ and $g \in \mathcal{G}$, let the algorithm $IWERM(\mathcal{H}, S_{s2}; g)$ can output a predictor $\hat{h} \in \mathcal{H}$, for which*

$$\tilde{L}_{\mathcal{D}_s}(\hat{h}, g) - \tilde{L}_{\mathcal{D}_s}(h^\star, g) \leq \epsilon_2 (\mathcal{H}, S_{s2}, \delta; g) \tag{38}$$

*with probability at least $1 - \delta$.*

# B    Proof of Theorem 1 in the Main Manuscript

**Theorem 1.** *Suppose $\mathbb{E}_{z \sim \mathcal{D}_s}[\tilde{\ell}(h(x), g(x); z)] = \tilde{L}_{\mathcal{D}_s}(h, g)$ satisfies Assumptions 1 to 3. Let $\hat{g}$ be the output of $DRE(\mathcal{G}, S_{s1}, S_t)$. We then have:*

*If $\beta_2$ in Item c of Assumption 3 is $0$, then the Algorithm 1 will produce a predictor $\hat{h}$ such that, with probability at least $1 - \delta$,*

$$\tilde{L}_{\mathcal{D}_s}(\hat{h}, g_0) - \tilde{L}_{\mathcal{D}_s}(h^\star, g_0) \leq \frac{\beta_1 \kappa}{2\lambda} \epsilon_1 \left( \mathcal{G}, S_{s1}, S_t, \frac{\delta}{2} \right)^{2k_2} + \frac{\beta_1}{\lambda} \epsilon_2 \left( \mathcal{H}, S_{s2}, \frac{\delta}{2}; \hat{g} \right) \tag{39}$$

*If $\beta_2$ in Item c of Assumption 3 is $> 0$, then the Algorithm 1 will produce a predictor $\hat{h}$ such that, with probability at least $1 - \delta$,*

$$\tilde{L}_{\mathcal{D}_s}(\hat{h}, g_0) - \tilde{L}_{\mathcal{D}_s}(h^\star, g_0) \leq \left( \frac{\beta_1 \kappa}{\lambda} + \frac{\beta_1 \beta_2^2}{2\lambda^2} \right) \epsilon_1 \left( \mathcal{G}, S_{s1}, S_t, \frac{\delta}{2} \right)^{2k_2} + \frac{2\beta_1}{\lambda} \epsilon_2 \left( \mathcal{H}, S_{s2}, \frac{\delta}{2}; \hat{g} \right) \tag{40}$$

Since our proof repeatedly invokes Taylor's theorem, we first review it and provide its directional derivative generalization version.

**Proposition 1** (Taylor expansion). *Let $a \leq b$ be fixed and let $f : I \to \mathcal{R}$, where $I \subseteq \mathcal{R}$ is an open interval containing $a, b$. If $f$ is $(k+1)$-times differentiable, then there exists $c \in [a, b]$ such that*

$$f(a) = f(b) + \sum_{i=1}^{k} \frac{1}{i!} f^{(i)}(b)(a-b)^i + \frac{1}{(k+1)!} f^{(k+1)}(c)(a-b)^{k+1} \tag{41}$$

*Let $F : \mathcal{F} \to \mathcal{R}$, where $\mathcal{F}$ is a vector space of functions. For any $g, g' \in \mathcal{F}$, if $t \mapsto F(t \cdot g + (1-t) \cdot g')$ is $(k+1)$-times differentiable over an open interval containing $[0, 1]$, then there exists $\bar{g} \in \mathrm{conv}(\{g, g'\})$ such that*

$$F(g') = F(g) + \sum_{i=1}^{k} \frac{1}{i!} \partial_g^i F(g) \underbrace{[g'-g, \ldots, g'-g]}_{i \ times} + \frac{1}{(k+1)!} \partial_g^{k+1} F(\bar{g}) \underbrace{[g'-g, \ldots, g'-g]}_{k+1 \ times}. \tag{42}$$

**Remark 5.** *The Gateaux derivative is a one-dimensional calculation along a specified direction. Because it's one-dimensional, we can use ordinary one-dimensional calculus to compute it. Thus, it is usually easy to compute a Gateaux differential even when the space $\mathcal{F}$ is infinite dimensional. The Taylor expansion for the Gateaux derivative can be obtained by applying a standard Taylor expansion to the function of the single real variable $T(t) = F(t \cdot g + (1-t) \cdot g')$, where $t \in [0, 1]$.*

*Proof of Theorem 1.* We start with a second-order Taylor expansion on the risk at $g_0$ to decompose the excess risk into two components. There exists $\bar{h} \in \mathrm{star}(\mathcal{H}, h^\star)$ such that

$$\tilde{L}_{\mathcal{D}_s}(\hat{h}, g_0) - \tilde{L}_{\mathcal{D}_s}(h^\star, g_0) = \partial_h \tilde{L}_{\mathcal{D}_s}(h^\star, g_0)[\hat{h} - h^\star] + \frac{1}{2} \partial_h^2 \tilde{L}_{\mathcal{D}_s}(\bar{h}, g_0)[\hat{h} - h^\star, \hat{h} - h^\star] \tag{43}$$

By Assumption 2, we know the first part in Eq. (43) is non-negative. So we defer the focus on this item. Additionally, by Item b in Assumption 3, we conclude

$$\partial_h^2 \tilde{L}_{\mathcal{D}_s}(\bar{h}, g_0)[\hat{h} - h^\star, \hat{h} - h^\star] \leq \beta_1 \|\hat{h} - h^\star\|_{\mathcal{H}}^{2k_1} \tag{44}$$

Combining Eq. (43) and Eq. (44) leads to the following inequality:

$$\tilde{L}_{\mathcal{D}_s}(\hat{h}, g_0) - \tilde{L}_{\mathcal{D}_s}(h^\star, g_0) \leq \partial_h \tilde{L}_{\mathcal{D}_s}(h^\star, g_0)[\hat{h} - h^\star] + \frac{\beta_1}{2} \|\hat{h} - h^\star\|_{\mathcal{H}}^{2k_1} \tag{45}$$

We shall bound $\|\hat{h} - h^\star\|_{\mathcal{H}}^{2k_1}$ first. Using a second-order Taylor expansion on the risk at $\hat{g}$, there exists $\bar{h} \in \mathrm{star}(\mathcal{H}, h^\star)$ such that

$$\tilde{L}_{\mathcal{D}_s}(\hat{h}, \hat{g}) - \tilde{L}_{\mathcal{D}_s}(h^\star, \hat{g}) = \partial_h \tilde{L}_{\mathcal{D}_s}(h^\star, \hat{g})[\hat{h} - h^\star] + \frac{1}{2} \partial_h^2 \tilde{L}_{\mathcal{D}_s}(\bar{h}, \hat{g})[\hat{h} - h^\star, \hat{h} - h^\star] \tag{46}$$

Applying a second-order Taylor expansion to the first item of Eq. (46) again yields that there exists $\bar{g} \in \mathrm{star}(\mathcal{G}, g_0)$ such that

$$\begin{aligned}
\partial_h \tilde{L}_{\mathcal{D}_s}(h^\star, \hat{g})[\hat{h} - h^\star] &= \partial_h \tilde{L}_{\mathcal{D}_s}(h^\star, g_0)[\hat{h} - h^\star] + \partial_g \partial_h \tilde{L}_{\mathcal{D}_s}(h^\star, g_0)[\hat{h} - h^\star, \hat{g} - g_0] \\
&+ \frac{1}{2} \partial_g^2 \partial_h \tilde{L}_{\mathcal{D}_s}(h^\star, \bar{g})[\hat{h} - h^\star, \hat{g} - g_0, \hat{g} - g_0]
\end{aligned} \tag{47}$$

Combining these two Taylor expansions (replacing the first item of Eq. (46) with RHS of Eq. (47)) implies:

$$\begin{aligned}
\tilde{L}_{\mathcal{D}_s}(\hat{h}, \hat{g}) - \tilde{L}_{\mathcal{D}_s}(h^\star, \hat{g}) &= \partial_h \tilde{L}_{\mathcal{D}_s}(h^\star, g_0)[\hat{h} - h^\star] + \partial_g \partial_h \tilde{L}_{\mathcal{D}_s}(h^\star, g_0)[\hat{h} - h^\star, \hat{g} - g_0] \\
&+ \frac{1}{2} \partial_g^2 \partial_h \tilde{L}_{\mathcal{D}_s}(h^\star, \bar{g})[\hat{h} - h^\star, \hat{g} - g_0, \hat{g} - g_0] + \frac{1}{2} \partial_h^2 \tilde{L}_{\mathcal{D}_s}(\bar{h}, \hat{g})[\hat{h} - h^\star, \hat{h} - h^\star]
\end{aligned} \tag{48}$$

Eq. (48) is the main inequality to get the upper bound of $\|\hat{h} - h^\star\|_{\mathcal{H}}^{2k_1}$. For the second item of Eq. (48), Assumption 1 yields:

$$\partial_g \partial_h \tilde{L}_{\mathcal{D}_s}(h^\star, g_0)[\hat{h} - h^\star, \hat{g} - g_0] = 0 \tag{49}$$

For the fourth item of Eq. (48), Item a of Assumption 3 leads to:

$$\partial_h^2 \tilde{L}_{\mathcal{D}_s}\left(\bar{h}, \hat{g}\right)[\hat{h} - h^\star, \hat{h} - h^\star] \geq \lambda \|\hat{h} - h^\star\|_{\mathcal{H}}^{2k_1} - \kappa \|\hat{g} - g_0\|_{\mathcal{G}}^{2k_2}, \forall \bar{h} \in \text{star}\left(\mathcal{H}, h^\star\right). \tag{50}$$

For the third item of Eq. (48), Item c of Assumption 3 concludes:

$$-\frac{1}{2}\partial_g^2 \partial_h \tilde{L}_{\mathcal{D}_s}\left(h^\star, \bar{g}\right)[\hat{h} - h^\star, \hat{g} - g_0, \hat{g} - g_0] \leq \frac{\beta_2}{2}\|\hat{g} - g_0\|_{\mathcal{G}}^{k_2} \cdot \|\hat{h} - h^\star\|_{\mathcal{H}}^{k_1} \tag{51}$$

Let $d > 0$ be any constant, $a = \frac{\|\hat{g} - g_0\|_{\mathcal{G}}^{k_2}}{\sqrt{d}}$ and $b = \|\hat{h} - h^\star\|_{\mathcal{H}}^{k_1} \cdot \sqrt{d}$. By the Algorithmic Mean-Geometric Mean (AM-GM) inequality, namely, $\forall a, b \in \mathcal{R}, ab \leq \frac{a^2 + b^2}{2}$:

$$\|\hat{g} - g_0\|_{\mathcal{G}}^{k_2} \cdot \|\hat{h} - h^\star\|_{\mathcal{H}}^{k_1} = ab \leq \frac{a^2 + b^2}{2} = \frac{1}{2}\left(\frac{\|\hat{g} - g_0\|_{\mathcal{G}}^{2k_2}}{d} + \|\hat{h} - h^\star\|_{\mathcal{H}}^{2k_1} \cdot d\right) \tag{52}$$

Based on Eqs. (48) to (52), we get:

$$\|\hat{h} - h^\star\|_{\mathcal{H}}^{2k_1} \leq \frac{\kappa}{\lambda}\|\hat{g} - g_0\|_{\mathcal{G}}^{2k_2} + \frac{2}{\lambda}\left(\tilde{L}_{\mathcal{D}_s}(\hat{h}, \hat{g}) - \tilde{L}_{\mathcal{D}_s}(h^\star, \hat{g})\right)$$
$$-\frac{2}{\lambda}\partial_h \tilde{L}_{\mathcal{D}_s}(h^\star, g_0)[\hat{h} - h^\star] + \frac{\beta_2}{2d\lambda}\|\hat{g} - g_0\|_{\mathcal{G}}^{2k_2} + \frac{\beta_2 d}{2\lambda}\|\hat{h} - h^\star\|_{\mathcal{H}}^{2k_1} \tag{53}$$

To combine like items, we need to discuss this inequality in two cases.

1. $\beta_2 = 0$   Replacing $\beta_2$ with 0 in Eq. (53):

$$\|\hat{h} - h^\star\|_{\mathcal{H}}^{2k_1} \leq \frac{\kappa}{\lambda}\|\hat{g} - g_0\|_{\mathcal{G}}^{2k_2} + \frac{2}{\lambda}\left(\tilde{L}_{\mathcal{D}_s}(\hat{h}, \hat{g}) - \tilde{L}_{\mathcal{D}_s}(h^\star, \hat{g})\right) - \frac{2}{\lambda}\partial_h \tilde{L}_{\mathcal{D}_s}(h^\star, g_0)[\hat{h} - h^\star] \tag{54}$$

Combining Eq. (45) and Eq. (54):

$$\tilde{L}_{\mathcal{D}_s}(\hat{h}, g_0) - \tilde{L}_{\mathcal{D}_s}(h^\star, g_0) \leq \frac{\beta_1 \kappa}{2\lambda}\|\hat{g} - g_0\|_{\mathcal{G}}^{2k_2} + \frac{\beta_1}{\lambda}\left(\tilde{L}_{\mathcal{D}_s}(\hat{h}, \hat{g}) - \tilde{L}_{\mathcal{D}_s}(h^\star, \hat{g})\right)$$
$$+ (1 - \frac{\beta_1}{\lambda})\partial_h \tilde{L}_{\mathcal{D}_s}(h^\star, g_0)[\hat{h} - h^\star] \tag{55}$$

By Assumption 2 and using the fact that $\frac{\beta_1}{\lambda} \geq 1$ without loss of generality, we get:

$$\tilde{L}_{\mathcal{D}_s}(\hat{h}, g_0) - \tilde{L}_{\mathcal{D}_s}(h^\star, g_0) \leq \frac{\beta_1 \kappa}{2\lambda}\|\hat{g} - g_0\|_{\mathcal{G}}^{2k_2} + \frac{\beta_1}{\lambda}\left(\tilde{L}_{\mathcal{D}_s}(\hat{h}, \hat{g}) - \tilde{L}_{\mathcal{D}_s}(h^\star, \hat{g})\right) \tag{56}$$

Lastly, we use the convergence rates assumed in Definitions 1 and 2:

$$\mathbb{P}_{S_{s1} \sim \mathcal{D}_s, S_t \sim \mathcal{D}_t}[\|\hat{g} - g_0\|_{\mathcal{G}} > \epsilon_1\left(\mathcal{G}, S_{s1}, S_t, \delta/2\right)] \leq \frac{\delta}{2}$$
$$\mathbb{P}_{S_{s2} \sim \mathcal{D}_s}[\tilde{L}_{\mathcal{D}_s}(\hat{h}, \hat{g}) - \tilde{L}_{\mathcal{D}_s}(h^\star, \hat{g}) > \epsilon_2(\mathcal{H}, S_{s2}, \frac{\delta}{2}; \hat{g})] \leq \frac{\delta}{2} \tag{57}$$

Since $S_{s1}, S_t, S_{s2}$ are mutually independent, we have

$$\mathbb{P}_{S_{s1} \sim \mathcal{D}_s, S_t \sim \mathcal{D}_t, S_{s2} \sim \mathcal{D}_s}\big[\|\hat{g} - g_0\|_{\mathcal{G}} > \epsilon_1\left(\mathcal{G}, S_{s1}, S_t, \delta/2\right) \cup$$
$$\tilde{L}_{\mathcal{D}_s}(\hat{h}, \hat{g}) - \tilde{L}_{\mathcal{D}_s}(h^\star, \hat{g}) > \epsilon_2(\mathcal{H}, S_{s2}, \frac{\delta}{2}; \hat{g})\big] \leq \frac{\delta}{2} + \frac{\delta}{2} = \delta \tag{58}$$

$$\tilde{L}_{\mathcal{D}_s}(\hat{h}, g_0) - \tilde{L}_{\mathcal{D}_s}(h^\star, g_0) \leq \frac{\beta_1 \kappa}{2\lambda}\epsilon_1\left(\mathcal{G}, S_{s1}, S_t, \delta/2\right)^{2k_2} + \frac{\beta_1}{\lambda}\epsilon_2(\mathcal{H}, S_{s2}, \frac{\delta}{2}; \hat{g}) \tag{59}$$

with probability at least $1 - \delta$.

2. $\beta_2 \neq 0$

Let $d = \frac{\lambda}{\beta_2}$. Then $\frac{\beta_2}{2d\lambda} = \frac{\beta_2^2}{2\lambda^2}$ and $\frac{\beta_2 d}{2\lambda} = \frac{1}{2}$. Moving items and combining same kind of items in Eq. (53):

$$\|\hat{h} - h^\star\|_{\mathcal{H}}^{2k_1} \leq \left(\frac{2\kappa}{\lambda} + \frac{\beta_2^2}{\lambda^2}\right) \|\hat{g} - g_0\|_{\mathcal{G}}^{2k_2} + \frac{4}{\lambda}\left(\tilde{L}_{\mathcal{D}_s}(\hat{h}, \hat{g}) - \tilde{L}_{\mathcal{D}_s}(h^\star, \hat{g})\right) - \frac{2}{\lambda}\partial_h \tilde{L}_{\mathcal{D}_s}(h^\star, g_0)[\hat{h} - h^\star] \tag{60}$$

Analogous the above case, we conclude the following inequality by combining Eq. (45) and Eq. (60):

$$\tilde{L}_{\mathcal{D}_s}(\hat{h}, g_0) - \tilde{L}_{\mathcal{D}_s}(h^\star, g_0) \leq \left(\frac{\beta_1 \kappa}{\lambda} + \frac{\beta_1 \beta_2^2}{2\lambda^2}\right) \|\hat{g} - g_0\|_{\mathcal{G}}^{2k_2}$$
$$+ \frac{2\beta_1}{\lambda}\left(\tilde{L}_{\mathcal{D}_s}(\hat{h}, \hat{g}) - \tilde{L}_{\mathcal{D}_s}(h^\star, \hat{g})\right) + \left(1 - \frac{\beta_1}{\lambda}\right)\partial_h \tilde{L}_{\mathcal{D}_s}(h^\star, g_0)[\hat{h} - h^\star] \tag{61}$$

Likewise, by Assumption 2 and using the fact that $\frac{\beta_1}{\lambda} \geq 1$ without loss of generality, we have:

$$\tilde{L}_{\mathcal{D}_s}(\hat{h}, g_0) - \tilde{L}_{\mathcal{D}_s}(h^\star, g_0) \leq \left(\frac{\beta_1 \kappa}{\lambda} + \frac{\beta_1 \beta_2^2}{2\lambda^2}\right) \|\hat{g} - g_0\|_{\mathcal{G}}^{2k_2} + \frac{2\beta_1}{\lambda}\left(\tilde{L}_{\mathcal{D}_s}(\hat{h}, \hat{g}) - \tilde{L}_{\mathcal{D}_s}(h^\star, \hat{g})\right) \tag{62}$$

Using the assumed rates again, we get the desired result:

$$\tilde{L}_{\mathcal{D}_s}(\hat{h}, g_0) - \tilde{L}_{\mathcal{D}_s}(h^\star, g_0) \leq \left(\frac{\beta_1 \kappa}{\lambda} + \frac{\beta_1 \beta_2^2}{2\lambda^2}\right) \epsilon_1\left(\mathcal{G}, S_{s1}, S_t, \delta/2\right)^{2k_2} + \frac{2\beta_1}{\lambda}\epsilon_2(\mathcal{H}, S_{s2}, \frac{\delta}{2}; \hat{g}) \tag{63}$$

with probability at least $1 - \delta$.

$\square$

## C  Proof of Lemma 1 in the Main Manuscript

**Lemma 1.** *Suppose* $\mathbb{E}[y|x] = h^\star(x)$. *Then* $\tilde{\ell}(\zeta, \gamma; z) = (\zeta - y)^2 \cdot \gamma$ *has the following properties:*

1. $\mathbb{E}[\nabla_\zeta \nabla_\gamma \tilde{\ell}(h^\star(x), g_0(x); z)|x] = 0$

2. $\mathbb{E}[\nabla_\zeta \tilde{\ell}(h^\star(x), g_0(x); z)(h(x) - h^\star(x))] = 0$
   $\forall h \in \text{star}(\mathcal{H}, h^\star)$

3. $\mathbb{E}[\nabla_{\gamma\gamma}^2 \nabla_\zeta \ell(h^\star(x), \bar{g}(x); z)|x] = 0 \quad \forall g \in \text{star}(\mathcal{G}, g_0)$

*Proof of Lemma 1.* **Property 1**.

$$\nabla_\zeta \tilde{\ell}(h^\star(x), g_0(x); z) = g_0(x) \cdot 2 \cdot (h^\star(x) - y)$$
$$\nabla_\gamma \nabla_\zeta \tilde{\ell}(h^\star(x), g_0(x); z) = 2(h^\star(x) - y) \tag{64}$$
$$\mathbb{E}[\nabla_\zeta \nabla_\gamma \tilde{\ell}(h^\star(x), g_0(x); z)|x] = 2\mathbb{E}[(h^\star(x) - y)|x] = 2\mathbb{E}[h^\star(x) - \mathbb{E}[y|x]] = 0$$

, where the final formula follows from $h^\star(x) = \mathbb{E}[y|x]$.

**Property 2**.

$$\mathbb{E}[\nabla_\zeta \tilde{\ell}(h^\star(x), g_0(x); z)(h(x) - h^\star(x))]$$
$$= \mathbb{E}[g_0(x) \cdot 2(h^\star(x) - y) \cdot (h(x) - h^\star(x))]$$
$$= 2\mathbb{E}[\mathbb{E}[g_0(x) \cdot (h^\star(x) - y) \cdot (h(x) - h^\star(x))|x]] \tag{65}$$
$$= 2\mathbb{E}[g_0(x) \cdot (h(x) - h^\star(x))\mathbb{E}[(h^\star(x) - y)|x]] = 0$$

**Property 3**.

$$\nabla_\zeta \tilde{\ell}\left(h^\star\left(x\right), g\left(x\right); z\right) = g\left(x\right) \cdot 2\left(h^\star\left(x\right) - y\right)$$
$$\nabla_\gamma \nabla_\zeta \tilde{\ell}\left(h^\star\left(x\right), g\left(x\right); z\right) = 2\left(h^\star\left(x\right) - y\right) \tag{66}$$
$$\nabla_\gamma^2 \nabla_\zeta \tilde{\ell}\left(h^\star\left(x\right), g\left(x\right); z\right) = 0$$

$\square$

## D    Proof of Theorem 2 in the Main Manuscript

**Theorem 2.** *Suppose* $\mathbb{E}[y|x] = h^\star(x)$, $\sup_{x \in \mathcal{X}, h \in \mathcal{H}}|h(x)| \leq M_h$, $\forall g \in \mathcal{G}, x \in \mathcal{X}, g(x) \geq \eta > 0$, $g_0(x) \geq \eta$. *Then* $\tilde{L}_{\mathcal{D}_s}(h, g) = \mathbb{E}[\tilde{\ell}(h, g; z)]$ *satisfies Assumptions 1 to 3, with* $\lambda = \frac{1}{4}$, $k_1 = 1$, $k_2 = 2$, $\kappa = \frac{M_h^2}{2\eta^3}$, $\beta_1 = 1$ *and* $\beta_2 = 0$.

*Proof of Theorem 2.*        1. Assumption 1

$$\partial_g \partial_h \tilde{L}_{\mathcal{D}_s}(h^\star, g_0)[h - h^\star, g - g_0]$$
$$= \mathbb{E}[(h(x) - h^\star(x)) \nabla_\zeta \nabla_\gamma \tilde{\ell}(h^\star(x), g_0(x); z) \cdot (g(x) - g_0(x))] \tag{67}$$
$$= \mathbb{E}[(h(x) - h^\star(x)) \mathbb{E}[\nabla_\zeta \nabla_\gamma \tilde{\ell}(h^\star(x), g_0(x); z) |x] \cdot (g(x) - g_0(x))]$$

From Lemma 1, we know that $\mathbb{E}[\nabla_\zeta \nabla_\gamma \tilde{\ell}(h^\star(x), g_0(x); z)|x] = 0$. Then, the above formula is zero. Assumption 1 is proved.

2. Assumption 2

$$\partial_h \tilde{L}_{\mathcal{D}_s}(h^\star, g_0)[h - h^\star] = \mathbb{E}[\nabla_\zeta \tilde{\ell}(h^\star(x), g_0(x); z)(h(x) - h^\star(x))] \tag{68}$$

By Lemma 1, we have $\mathbb{E}[\nabla_\zeta \tilde{\ell}(h^\star(x), g_0(x); z)(h(x) - h^\star(x))] \geq 0, \forall h \in \text{star}(\mathcal{H}, h^\star)$. Therefore, $\partial_h \tilde{L}_{\mathcal{D}_s}(h^\star, g_0)[h - h^\star] \geq 0$. Assumption 2 is satisfied.

3. Assumption 3

    (a) Item a

$$\forall h' \in \mathcal{H}, \quad \partial_h^2 \tilde{L}_{\mathcal{D}_s}(\bar{h}, g)[h - h', h - h']$$
$$= \partial_h \mathbb{E}[2(\bar{h}(x) - y) g(x) (h(x) - h'(x))][h - h'] = \mathbb{E}[2g(x) \cdot (h(x) - h'(x))^2] \tag{69}$$

Therefore,

$$\partial_h^2 \tilde{L}_{\mathcal{D}_s}(\bar{h}, g)[h - h^\star, h - h^\star] = \mathbb{E}[2g(x) \cdot (h(x) - h^\star(x))^2] \tag{70}$$

Let $a = \sqrt{2g(x)}(h(x) - h^\star(x))$, $b = (\sqrt{2g_0(x)} - \sqrt{2g(x)})(h(x) - h^\star(x))$. Then $a + b = \sqrt{2g_0(x)}(h(x) - h^\star(x))$. By AM-GM inequality, we have

$$2g(x)(h(x) - h^\star(x))^2 = a^2 \geq \frac{1}{2}(a + b)^2 - b^2$$
$$= \frac{1}{2}(\sqrt{2g_0(x)}(h(x) - h^\star(x)))^2 - (\sqrt{2g_0(x)} - \sqrt{2g(x)})^2(h(x) - h^\star(x))^2 \tag{71}$$

Combining Eq. (70) and Eq. (71) leads to:

$$\partial_h^2 \tilde{L}_{\mathcal{D}_s}(\bar{h}, g)[h - h^\star, h - h^\star]$$
$$\geq \frac{1}{2}\|h - h^\star\|_{\mathcal{H}}^2 - \mathbb{E}[(\sqrt{2g_0(x)} - \sqrt{2g(x)})^2(h(x) - h^\star(x))^2] \tag{72}$$

Our following task is to lower bound the second part of Eq. (72).

$$(\sqrt{2g_0(x)} - \sqrt{2g(x)})^2 = \left(\frac{|2g_0(x) - 2g(x)|}{\sqrt{2g_0(x)} + \sqrt{2g(x)}}\right)^2 \leq \frac{1}{2\eta}(g_0(x) - g(x))^2 \tag{73}$$

Multiplying $(h(x) - h^\star(x))^2$ by both sides of Eq. (73) implies:

$$(\sqrt{2g_0(x)} - \sqrt{2g(x)})^2(h(x) - h^\star(x))^2 \leq \frac{1}{2\eta}(g_0(x) - g(x))^2(h(x) - h^\star(x))^2 \tag{74}$$

Let $c > 0$ be any number. $a = \frac{1}{2\eta\sqrt{c}}(g_0(x) - g(x))^2$, $b = \sqrt{c}(h(x) - h^\star(x))^2$. AM-GM inequality yields:

$$\begin{aligned}
\frac{1}{2\eta}(g_0(x) - g(x))^2(h(x) - h^\star(x))^2 &= \frac{1}{2\eta\sqrt{c}}(g_0(x) - g(x))^2 \cdot \sqrt{c}(h(x) - h^\star(x))^2 \\
&= ab \leq \frac{a^2 + b^2}{2} = \frac{\frac{1}{4\eta^2 c}(g_0(x) - g(x))^4 + c(h(x) - h^\star(x))^4}{2} \\
&= \frac{(g_0(x) - g(x))^4}{8\eta^2 c} + \frac{c}{2}(h(x) - h^\star(x))^4
\end{aligned} \tag{75}$$

Combining Eq. (74) and Eq. (75) leads to the following inequality:

$$\begin{aligned}
&\mathbb{E}[(\sqrt{2g_0(x)} - \sqrt{2g(x)})^2(h(x) - h^\star(x))^2] \\
&\leq \mathbb{E}\Big[\frac{(g_0(x) - g(x))^4}{8\eta^2 c} + \frac{c}{2}(h(x) - h^\star(x))^4\Big] \\
&= \frac{1}{8\eta^2 c}\|g - g_0\|_{\mathcal{G}}^4 + \frac{c}{2}\mathbb{E}[(h(x) - h^\star(x))^4]
\end{aligned} \tag{76}$$

Therefore, it suffices to upper bound $\mathbb{E}[(h(x) - h^\star(x))^4]$ using $\|h - h^\star\|_{\mathcal{H}}^2$.

$$\begin{aligned}
|h(x) - h^\star(x)| &\leq |h(x)| + |h^\star(x)| \leq 2M_h \\
|h(x) - h^\star(x)|^2 &\leq 4M_h^2 \\
|h(x) - h^\star(x)|^4 &\leq 4M_h^2(h(x) - h^\star(x))^2 \\
\mathbb{E}[(h(x) - h^\star(x))^4] &\leq 4M_h^2\mathbb{E}[(h(x) - h^\star(x))^2]
\end{aligned} \tag{77}$$

Since $\forall g \in \mathcal{G}, x \in \mathcal{X}, g(x) \geq \eta > 0$,

$$\frac{\mathbb{E}[2g_0(x) \cdot (h(x) - h^\star(x))^2]}{\mathbb{E}(h(x) - h^\star(x))^2} \geq 2\eta \tag{78}$$

Combining Eq. (72), Eq. (76), Eq. (77) and Eq. (78) yields:

$$\partial_h^2 \tilde{L}_{\mathcal{D}_s}(\bar{h}, g)[h - h^\star, h - h^\star] \geq (\frac{1}{2} - \frac{cM_h^2}{\eta})\|h - h^\star\|_{\mathcal{H}}^2 - \frac{1}{8\eta^2 c}\|g - g_0\|_{\mathcal{G}}^4 \tag{79}$$

Let $c = \frac{\eta}{4M_h^2}$. Then, $\frac{cM_h^2}{\eta} = \frac{1}{4}$, $\frac{1}{8\eta^2 c} = \frac{M_h^2}{2\eta^3}$

$$\partial_h^2 \tilde{L}_{\mathcal{D}_s}(\bar{h}, g)[h - h^\star, h - h^\star] \geq \frac{1}{4}\|h - h^\star\|_{\mathcal{H}}^2 - \frac{M_h^2}{2\eta^3}\|g - g_0\|_{\mathcal{G}}^4 \tag{80}$$

Item a is proved with corresponding constants $\lambda = \frac{1}{4}$, $\kappa = \frac{M_h^2}{2\eta^3}$

(b) Item b

By Eq. (69), we get:

$$\begin{aligned}
\partial_h^2 \tilde{L}_{\mathcal{D}_s}(\bar{h}, g)[h - h^\star, h - h^\star] &= \mathbb{E}[2g(x) \cdot (h(x) - h^\star(x))^2] \\
&= \|h - h^\star\|_{\mathcal{H}}^2 \leq 1 \cdot \|h - h^\star\|_{\mathcal{H}}^2
\end{aligned} \tag{81}$$

Item b is satisfied.

(c) Item c

For all $h \in \text{star}(\mathcal{H}, h^\star)$, $g \in \mathcal{G}$, and $\bar{g} \in \text{star}(\mathcal{G}, g_0)$:

$$\partial_g^2 \partial_h \tilde{L}_{\mathcal{D}_s}(h^\star, \bar{g})[h\text{–}h^\star, g\text{–}g_0, g\text{–}g_0] = \mathbb{E}[(h(x) - h^\star(x))(g(x) - g_0(x)) \tag{82}$$
$$\mathbb{E}[\nabla_{\gamma\gamma}^2 \nabla_\zeta \ell(h^\star(x), \bar{g}(x); z)|x](g(x) - g_0(x))]$$

Let $\Sigma(X) = \mathbb{E}[\nabla_{\gamma\gamma}^2 \nabla_\zeta \ell(h^\star(x), \bar{g}(x); z)|x]$

$$|\partial_g^2 \partial_h \tilde{L}_{\mathcal{D}_s}(h^\star, \bar{g})[h - h^\star, g - g_0, g - g_0]|$$
$$= |\mathbb{E}[(h(x)\text{–}h^\star(x))(g(x)\text{–}g_0(x))\Sigma(X)(g(x)\text{–}g_0(x))]| \tag{83}$$
$$\leq \mathbb{E}[|h(x)\text{–}h^\star(x)| \cdot |g(x)\text{–}g_0(x)|\Sigma(X)(g(x)\text{–}g_0(x))]$$

By Lemma 1, we know: $\forall g \in \text{star}(\mathcal{G}, g_0), \Sigma(X) = 0$.

$$|\partial_g^2 \partial_h \tilde{L}_{\mathcal{D}_s}(h^\star, \bar{g})[h\text{–}h^\star, g\text{–}g_0, g\text{–}g_0]| \leq 0 \cdot \mathbb{E}[|h(x)\text{–}h^\star(x)||g(x)\text{–}g_0(x)|^2] \tag{84}$$

Using Cauchy-Schwartz Inequality, i.e., $\mathbb{E}[|AB|] \leq \sqrt{\mathbb{E}[|A|^2]}\sqrt{\mathbb{E}[|B|^2]}$ implies:

$$|\partial_g^2 \partial_h \tilde{L}_{\mathcal{D}_s}(h^\star, \bar{g})[h\text{–}h^\star, g\text{–}g_0, g\text{–}g_0]| \leq 0 \cdot \mathbb{E}[|h(x)\text{–}h^\star(x)||g(x)\text{–}g_0(x)|^2] \tag{85}$$
$$\leq 0 \cdot \sqrt{\mathbb{E}[|h(x)\text{–}h^\star(x)|^2]}\sqrt{\mathbb{E}[|g(x)\text{–}g_0(x)|^4]} = 0 \cdot \|h\text{–}h^\star\|_{L_2(\mathcal{D}_s^x)} \cdot \|g\text{–}g_0\|_{\mathcal{G}}^2$$

By Eq. (78), we tell $\|h - h^\star\|_{L_2(\mathcal{D}_s^x)} \leq \frac{1}{\sqrt{2\eta}}\|h - h^\star\|_{\mathcal{H}}$.

$$|\partial_g^2 \partial_h \tilde{L}_{\mathcal{D}_s}(h^\star, \bar{g})[h - h^\star, g\text{–}g_0, g\text{–}g_0]| \leq 0 \cdot \|h - h^\star\|_{\mathcal{H}} \cdot \|g - g_0\|_{\mathcal{G}}^2 \tag{86}$$

Item c is proved.

In summary, there exist constants $k_1 = 1, k_2 = 2$ such that the conditions in Assumption 3 holds.

$\square$

# E  Proof of Theorem 3 in the Main Manuscript

**Theorem 3.** *Let $\mathcal{H} = \{x \mapsto \text{sig}(\langle w, x \rangle) : \|w\|_2 \leq 1\}$ and $\ell(h(x), y) = (h(x) - y)^2$. Suppose $\forall x \in \mathcal{X}, \|x\|_2 \leq B$ and $\forall g \in \mathcal{G}, \sup_{x \in X} g(x) \leq M$. When given $S_{s2} \sim \mathcal{D}_s$ and any $g \in \mathcal{G}$, IWERM($\mathcal{H}, S_{s2}; g$) outputs a predictor $\hat{h} \in \mathcal{H}$ for which*

$$\tilde{L}_{\mathcal{D}_s}(\hat{h}, g) - \tilde{L}_{\mathcal{D}_s}(h^\star, g)$$
$$\leq 8M \inf_{\Delta \in [0,D], N_X(\Delta; \mathcal{F}) \geq 10} \left\{ 2\Delta + \frac{2B\sqrt{d\log\left(1 + \frac{B}{2\Delta}\right)}}{\sqrt{n_2}} \right\} + 2M\sqrt{\frac{2\log\left(\frac{1}{\delta}\right)}{n_2}} \tag{87}$$

*where $D = \sup_{w,w' \in \mathbb{B}(0,1)} \sup_{x \in \mathcal{X}} |\text{sig}(\langle w, x \rangle) - \text{sig}(\langle w', x \rangle)|$ and $n_2$ is the size of $S_{s2}$.*

In this section, we provide the proof of Theorem 3 in detail. First, we state several key definitions and lemmas.

## E.1  Definitions

**Definition 4** (Rademacher average (Boucheron et al., 2005))**.** *Let $A \subset \mathcal{R}^n$ be a bounded set of vectors $a = (a_1, \ldots, a_n)$; the Rademacher average associated with $A$ is denoted as:*

$$\mathfrak{R}_n(A) = \mathbb{E}_\sigma[\sup_{a \in A} \frac{1}{n}|\sum_{i=1}^n \sigma_i a_i|]$$

*, where $\sigma = (\sigma_i, \ldots, \sigma_n)$ is the Rademacher vector, namely, $\sigma_1, \ldots, \sigma_n$ are independent uniform $\{\pm 1\}$.*

**Definition 5** (Sub-Gaussian random variable (Martin J., 2019)). *A random variable $V$ with mean $\mu = \mathbb{E}[V]$ is sub-Gaussian if there is a positive number of $\iota$ such that*

$$\mathbb{E}[e^{\lambda(V-\mu)}] \leq e^{\frac{\iota^2 \lambda^2}{2}} \quad \forall \lambda \in \mathcal{R} \tag{88}$$

**Definition 6** (Sub-Gaussian process (Martin J., 2019)). *A collection of zero-mean random variables $\{X_f, f \in \mathcal{F}\}$ is a sub-Gaussian process with respect to a metric $\rho_X$ on $\mathcal{F}$ if*

$$\mathbb{E}\left[e^{\lambda\left(X_f - X_{f'}\right)}\right] \leq e^{\frac{\lambda^2 \rho_X^2(f,f')}{2}} \quad \text{for all } f, f' \in \mathcal{F}, \text{ and } \lambda \in \mathbb{R}$$

**Definition 7** (Covering number (Martin J., 2019)). *A $\delta$-cover of a set $\mathbb{T}$ with respect to a metric $\rho$ is a set $\{\theta^1, \ldots, \theta^N\} \subset \mathbb{T}$ such that for each $\theta \in \mathbb{T}$, there exists some $i \in \{1, \ldots, N\}$ such that $\rho\left(\theta, \theta^i\right) \leq \delta$. The $\delta$-covering number $N(\delta; \mathbb{T}, \rho)$ is the cardinality of the smallest $\delta$-cover.*

### E.2 Lemmas

**Lemma 6.** *A Rademacher random variable $\sigma_i$ is sub-Gaussian with parameter $\iota = 1$.*

*Proof of Lemma 6.* By taking expectations and using the power-series for the exponential, we obtain:

$$\mathbb{E}[e^{\lambda\sigma_i}] = \frac{1}{2}(e^{-\lambda} + e^{\lambda}) = \frac{1}{2}\left(\sum_{k=0}^{\infty} \frac{(-\lambda)^k}{k!} + \sum_{k=0}^{\infty} \frac{(\lambda)^k}{k!}\right) = \sum_{k=0}^{\infty} \frac{\lambda^{2k}}{(2k)!} \leq 1 + \sum_{k=1}^{\infty} \frac{\lambda^{2k}}{2^k k!} = e^{\frac{\lambda^2}{2}} \tag{89}$$

which shows that $\sigma_i$ is sub-Gaussian with parameter $\iota = 1$ as claimed. $\qquad\square$

**Lemma 7** (Proposition 5.17 of (Martin J., 2019)). *Let $\{X_f, f \in \mathcal{F}\}$ be a zero-mean sub-Gaussian process with respect to the metric $\rho_X$. Then, for any $\Delta \in [0, D]$ such that $N_X(\Delta; \mathcal{F}) \geq 10$, we have*

$$\mathbb{E}\left[\sup_{f,f' \in \mathcal{F}} (X_f - X_{f'})\right] \leq 2\mathbb{E}\left[\sup_{\substack{\gamma, \gamma' \in \mathcal{F} \\ \rho_X(\gamma, \gamma') \leq \Delta}} (X_\gamma - X_{\gamma'})\right] + 4\sqrt{D^2 \log N_X(\Delta; \mathcal{F})} \tag{90}$$

*where $D := \sup_{f,f' \in \mathcal{F}} \rho_X(f, f')$ denotes the diameter of $\mathcal{F}$.*

**Lemma 8.** *Let $\mathbb{B}(0,1)$ be the unit ball with respect to $\|\cdot\|_2$, and $N\left(\frac{4\Delta}{B}; \mathbb{B}(0,1), \|\cdot\|_2\right)$ be the $\frac{4\Delta}{B}$-covering number of $\mathbb{B}(0,1)$ with respect to the $\|\cdot\|_2$-metric. Then, $N_X(\Delta; \mathcal{F}) \leq N\left(\frac{4\Delta}{B}; \mathbb{B}(0,1), \|\cdot\|_2\right)$.*

*Proof of Lemma 8.* Assume $\{w_1, \ldots, w_n\}$ is a $\frac{4\Delta}{B}$-covering of $\mathbb{B}(0,1)$. We then have: $\forall w' \in \mathbb{B}(0,1), \exists w_i$ such that $\|w_i - w'\|_2 \leq \frac{4\Delta}{B}$. Then, $\forall f' = \text{sig}(\langle w', x\rangle) - y, \exists f_i = \text{sig}(\langle w_i, x\rangle) - y$ such that $\|f_i - f'\|_\infty \leq \frac{1}{4}\|w_i - w'\|_2 \cdot \|x\|_2 \leq \frac{1}{4} \cdot \frac{4\Delta}{B} \cdot B = \Delta$. Therefore, $N_X(\Delta; \mathcal{F}) \leq N\left(\frac{4\Delta}{B}; \mathbb{B}(0,1), \|\cdot\|_2\right)$. $\qquad\square$

**Lemma 9.** $\log N\left(\frac{4\Delta}{B}; \mathbb{B}(0,1), \|\cdot\|_2\right) \leq d\log\left(1 + \frac{B}{2\Delta}\right)$

*Proof of Lemma 9.* Example 5.8 of (Martin J., 2019) concludes $\log N(\Delta; \mathbb{B}(0,1), \|\cdot\|_2) \leq d\log\left(1 + \frac{2}{\Delta}\right)$. Therefore, $\log N\left(\frac{4\Delta}{B}; \mathbb{B}(0,1), \|\cdot\|_2\right) \leq d\log\left(1 + \frac{B}{2\Delta}\right)$. $\qquad\square$

**Lemma 10.** $\phi(t) = t^2$ is $\rho$-Lipschitz over the set $C = \{t : |t| \leq \frac{\rho}{2}\}$.

*Proof of Lemma 10.* For any $t_1, t_2 \in C$, $|t_1^2 - t_2^2| = |t_1 + t_2||t_1 - t_2| \leq 2 \cdot \frac{\rho}{2}|t_1 - t_2| = \rho|t_1 - t_2|$ $\qquad\square$

**Lemma 11** (Lemma 1 of (Vogel et al., 2020)). *When given a sample $z_1, \ldots, z_n$ from distribution $\mathcal{D}_s$ and any $g \in \mathcal{G}$, IWERM($\mathcal{H}, S_{s2}; g$) outputs a predictor $\hat{h} \in \mathcal{H}$, for which*

$$
\tilde{L}_{\mathcal{D}_s}(\hat{h}, g) - \tilde{L}_{\mathcal{D}_s}(h^\star, g) \leq 4\|g\|_\infty \mathbb{E}_{z_1, \ldots, z_n \sim \mathcal{D}_s}\left[\mathbb{E}_\sigma[\sup_{h \in \mathcal{H}} \frac{1}{n} |\sum_{i=1}^n \sigma_i \ell(h, y_i)|]\right]
$$
$$
+ 2\|g\|_\infty \sup_{(h,z) \in \mathcal{H} \times \mathcal{Z}} \ell(h, y) \sqrt{\frac{2 \log\left(\frac{1}{\delta}\right)}{n}}
$$

(91)

*with probability at least $1 - \delta$.*

*Proof of Theorem 3.* Lemma 11 has given an upper bound of the estimation error except that the Rademacher complexity item is not well-bounded. Therefore, it suffices to bound that item.

Let $A = \{(h(x_1) - y_1, h(x_2) - y_2, \ldots, h(x_n) - y_n) : h \in \mathcal{H}\}$, $f(z) = h(x) - y$, and $a_i = h(x_i) - y_i = f(z_i)$. Let $\phi(t) = t^2$. Then, $\phi \circ A = \{(a_1^2, \ldots, a_n^2) : h \in \mathcal{H}\}$. $\mathbb{E}_\sigma[\sup_{h \in \mathcal{H}} \frac{1}{n}|\sum_{i=1}^n \sigma_i \ell(h, y_i)|] = \Re_n(\phi \circ A)$. The contraction principle ((Ledoux & Talagrand, 1991)) states that if $\phi : \mathcal{R} \to \mathcal{R}$ is a function with $\phi(0) = 0$ and Lipschitz constant $L_\phi$ and $\phi \circ A$ is the set of vectors of form $(\phi(a_1), \ldots, \phi(a_n)) \in \mathcal{R}^n$ with $(a_1, \ldots, a_n) \in A$, then $\Re_n(\phi \circ A) \leq L_\phi \Re_n(A)$. This principle plus the fact that $\phi(t) = t^2$ is 2-Lipschitz (Lemma 10) yields: $\Re_n(\phi \circ A) \leq 2\Re_n(A) = 2\mathbb{E}_\sigma\left[\sup_{f \in \mathcal{F}} \frac{1}{n}|\sum_{i=1}^n \sigma_i f(z_i)|\right]$, where $\mathcal{F} = \{f : f(z) = h(x) - y, h \in \mathcal{H}\}$. So bounding $\Re_n(A)$ is our following goal.

Let $X_f = \frac{1}{\sqrt{n}} \sum_{i=1}^n \sigma_i f(z_i)$ and $\widetilde{\mathcal{F}} = \mathcal{F} \cup -\mathcal{F}$. We then have

$$
\Re_n(A) = \mathbb{E}_\sigma\left[\sup_{f \in \mathcal{F}} \frac{1}{n}\left|\sum_{i=1}^n \sigma_i f(z_i)\right|\right] = \mathbb{E}_\sigma\left[\sup_{f \in \widetilde{\mathcal{F}}} \frac{1}{n}\sum_{i=1}^n \sigma_i f(z_i)\right]
$$
$$
= \mathbb{E}_\sigma\left[\sup_{f \in \widetilde{\mathcal{F}}} \frac{1}{\sqrt{n}} X_f\right] = \frac{1}{\sqrt{n}}\mathbb{E}_\sigma\left[\sup_{f \in \widetilde{\mathcal{F}}} X_f\right] \leq \frac{1}{\sqrt{n}}\mathbb{E}_\sigma\left[\sup_{f, f' \in \widetilde{\mathcal{F}}} X_f - X_f'\right]
$$

(92)

It is easy to prove that $\{X_f, f \in \mathcal{F}\}$ is a sub-Gaussian process[2] with respect to the metric $\rho_X(f, f') = \|f - f'\|_\infty$, whose proof is as follows:

$$
\mathbb{E}_\sigma[e^{\lambda(X_f - X_{f'})}] = \mathbb{E}_\sigma[e^{\frac{1}{\sqrt{n}}\lambda \sum_{i=1}^n \sigma_i(f(z_i) - f'(z_i))}] = \Pi_{i=1}^n \mathbb{E}_{\sigma_i}[e^{\frac{1}{\sqrt{n}}\lambda \sigma_i(f(z_i) - f'(z_i))}]
$$
$$
\leq \Pi_{i=1}^n \mathbb{E}_{\sigma_i}[e^{\frac{\lambda^2 (f(z_i) - f'(z_i))^2}{2n}}] \quad \text{(Lemma 6)} \leq \Pi_{i=1}^n \mathbb{E}_{\sigma_i}[e^{\frac{\lambda^2 \|f - f'\|_\infty^2}{2n}}] = e^{\frac{\lambda^2 \|f - f'\|_\infty^2}{2}}
$$

(93)

---

[2]Definition 6

Let $D = \sup_{f,f' \in \mathcal{F}} \rho_X(f, f')$. By Lemma 7, for any $\Delta \in [0, D]$, we have:

$$
\begin{aligned}
\mathbb{E}_\sigma \left[ \sup_{f, f' \in \widetilde{\mathcal{F}}} X_f - X_f' \right] &\leq 2\mathbb{E}_\sigma \left[ \sup_{\substack{\gamma, \gamma' \in \mathcal{F} \\ \rho_X(\gamma, \gamma') \leq \Delta}} (X_\gamma - X_{\gamma'}) \right] + 4\sqrt{D^2 \log N_X(\Delta; \mathcal{F})} \\
&= 2\mathbb{E}_\sigma \left[ \sup_{\substack{\gamma, \gamma' \in \mathcal{F} \\ \|\gamma - \gamma'\|_\infty \leq \Delta}} \left( \frac{1}{\sqrt{n}} \sum_{i=1}^n \sigma_i \left( \gamma(z_i) - \gamma'(z_i) \right) \right) \right] + 4\sqrt{D^2 \log N_X(\Delta; \mathcal{F})} \\
&\leq 2\mathbb{E}_\sigma \left[ \sup_{\substack{\gamma, \gamma' \in \mathcal{F} \\ \|\gamma - \gamma'\|_\infty \leq \Delta}} \left( \frac{1}{\sqrt{n}} \sum_{i=1}^n |\sigma_i| |\gamma(z_i) - \gamma'(z_i)| \right) \right] + 4\sqrt{D^2 \log N_X(\Delta; \mathcal{F})} \\
&\leq 2\mathbb{E}_\sigma \left[ \frac{\Delta}{\sqrt{n}} \sum_{i=1}^n |\sigma_i| \right] + 4\sqrt{D^2 \log N_X(\Delta; \mathcal{F})} \leq 2\sqrt{n}\Delta + 4\sqrt{D^2 \log N_X(\Delta; \mathcal{F})} \\
&\leq 2\sqrt{n}\Delta + 4D\sqrt{\log N\left( \frac{4\Delta}{B}; \mathbb{B}(0,1), \|\cdot\|_2 \right)} \quad \text{(Lemma 8)} \\
&\leq 2\sqrt{n}\Delta + 4D\sqrt{d \log\left( 1 + \frac{B}{2\Delta} \right)} \quad \text{(Lemma 9)}
\end{aligned}
\tag{94}
$$

Moreover,

$$
\begin{aligned}
D &= \sup_{f,f' \in \mathcal{F}} \rho_X(f, f') = \sup_{f,f' \in \mathcal{F}} \sup_{(x,y) \in \mathcal{X} \times \mathcal{Y}} \left\{ |\text{sig}(\langle w, x \rangle) - y - \text{sig}(\langle w', x \rangle) + y| \right\} \\
&= \sup_{f,f' \in \mathcal{F}} \sup_{(x,y) \in \mathcal{X} \times \mathcal{Y}} \left\{ |\text{sig}(\langle w, x \rangle) - \text{sig}(\langle w', x \rangle)| \right\} \leq \sup_{f,f' \in \mathcal{F}} \sup_{(x,y) \in \mathcal{X} \times \mathcal{Y}} \left\{ \frac{1}{4} |\langle w, x \rangle - \langle w', x \rangle| \right\} \\
&= \sup_{f,f' \in \mathcal{F}} \sup_{(x,y) \in \mathcal{X} \times \mathcal{Y}} \left\{ \frac{1}{4} |\langle w - w', x \rangle| \right\} \leq \sup_{f,f' \in \mathcal{F}} \sup_{(x,y) \in \mathcal{X} \times \mathcal{Y}} \left\{ \frac{1}{4} \|w - w'\|_2 \|x\|_2 \right\} \leq \frac{B}{2}
\end{aligned}
\tag{95}
$$

Combining Eq. (94) and Eq. (95) yields: $\mathbb{E}_\sigma \left[ \sup_{f \in \widetilde{\mathcal{F}}} X_f \right] \leq 2\sqrt{n}\Delta + 2B\sqrt{d \log\left( 1 + \frac{B}{2\Delta} \right)}$. Since Eq. (94) holds for any $\Delta \in [0, D]$, we take the inf of the upper bound, namely

$$
\mathbb{E}_\sigma \left[ \sup_{f \in \widetilde{\mathcal{F}}} X_f \right] \leq \inf_{\Delta \in [0,D], N_X(\Delta; \mathcal{F}) \geq 10} \left\{ 2\sqrt{n}\Delta + 2B\sqrt{d \log\left( 1 + \frac{B}{2\Delta} \right)} \right\}
\tag{96}
$$

Combining Eq. (92) and Eq. (96), we obtain:

$$
\mathfrak{R}_n(A) = \frac{1}{\sqrt{n}} \mathbb{E}_\sigma \left[ \sup_{f \in \widetilde{\mathcal{F}}} X_f \right] \leq \frac{1}{\sqrt{n}} \inf_{\Delta \in [0,D], N_X(\Delta; \mathcal{F}) \geq 10} \left\{ 2\sqrt{n}\Delta + 2B\sqrt{d \log\left( 1 + \frac{B}{2\Delta} \right)} \right\}
\tag{97}
$$

Lastly, by Lemma 11, with probability at least $1 - \delta$,

$$
\begin{aligned}
&\tilde{L}_{\mathcal{D}_s}(\hat{h}, g) - \tilde{L}_{\mathcal{D}_s}(h^\star, g) \\
&\leq 4\|g\|_\infty \mathbb{E}_{z_1, \ldots, z_n \sim D_s} \left[ \mathfrak{R}_n(\phi \circ A) \right] + 2\|g\|_\infty \sup_{(h,z) \in \mathcal{H} \times \mathcal{Z}} \ell(h, y) \sqrt{\frac{2 \log(\frac{1}{\delta})}{n}} \\
&\leq 8\|g\|_\infty \inf_{\substack{\Delta \in [0,D], \\ N_X(\Delta; \mathcal{F}) \geq 10}} \left\{ 2\Delta + \frac{2B\sqrt{d \log(1 + \frac{B}{2\Delta})}}{\sqrt{n}} \right\} + 2\|g\|_\infty \sup_{(h,z) \in \mathcal{H} \times \mathcal{Z}} \ell(h, y) \sqrt{\frac{2 \log(\frac{1}{\delta})}{n}} \\
&\leq 8M \inf_{\substack{\Delta \in [0,D], \\ N_X(\Delta; \mathcal{F}) \geq 10}} \left\{ 2\Delta + \frac{2B\sqrt{d \log\left( 1 + \frac{B}{2\Delta} \right)}}{\sqrt{n}} \right\} + 2M\sqrt{\frac{2 \log\left( \frac{1}{\delta} \right)}{n}} \text{ (by } \sup_{x \in \mathcal{X}} g(x) \leq M\text{)}
\end{aligned}
\tag{98}
$$

where $D = \sup_{w,w' \in \mathbb{B}(0,1)} \sup_{x \in \mathcal{X}} |\text{sig}(\langle w, x \rangle) - \text{sig}(\langle w', x \rangle)|$ $\qquad \square$

# F    Proof of Lemma 3 in the Main Manuscript

**Lemma 3.** *Suppose* $\forall g \in \mathcal{G}, \sup_{x \in \mathcal{X}} g(x) \leq M, \ g_0(x) \leq \eta_1.$ *Then* $\|\hat{g} - g_0\|_{L_4(\mathcal{D}_s^x)} \leq 2^{\frac{1}{4}} (\eta_1 + M)^{\frac{3}{4}} h_{p_s}(\hat{g}, g_0)^{\frac{1}{2}}.$

*Proof of Lemma 3.*

$$
\begin{aligned}
h_{p_s}(\hat{g}, g_0)^2 &= \int_{\mathcal{X}} \left(\sqrt{\hat{g}(x)} - \sqrt{g_0(x)}\right)^2 p_s(x)\, dx = \int_{\mathcal{X}} \frac{(\hat{g}(x) - g_0(x))^2}{\left(\sqrt{\hat{g}(x)} + \sqrt{g_0(x)}\right)^2} p_s(x) dx \\
&\geq \int_{\mathcal{X}} \frac{(\hat{g}(x) - g_0(x))^2}{\left(\sqrt{M} + \sqrt{\eta_1}\right)^2} p_s(x)\, dx \geq \frac{1}{2(M + \eta_1)} \|\hat{g} - g_0\|_{L_2(\mathcal{D}_s^x)}^2
\end{aligned} \tag{99}
$$

$$
\begin{aligned}
\|\hat{g} - g_0\|_{L_4(\mathcal{D}_s^x)}^4 &= \int_{\mathcal{X}} (\hat{g}(x) - g_0(x))^4 p_s(x)\, dx \\
&= (\eta_1 + M)^4 \int_{\mathcal{X}} \left(\frac{\hat{g}(x) - g_0(x)}{\eta_1 + M}\right)^4 p_s(x)\, dx \leq (\eta_1 + M)^4 \int_{\mathcal{X}} \left(\frac{\hat{g}(x) - g_0(x)}{\eta_1 + M}\right)^2 p_s(x) dx \\
&= (\eta_1 + M)^2 \|\hat{g} - g_0\|_{L_2(\mathcal{D}_s^x)}^2 \leq 2(\eta_1 + M)^3 h_{p_s}(\hat{g}, g_0)^2
\end{aligned} \tag{100}
$$

Both sides take the $1/4$ power yields the statement of the lemma. $\qquad \square$

# G    Proof of Lemma 5 in the Main Manuscript

**Lemma 5.** *Suppose* $\forall g \in \mathcal{G}, \sup_{x \in \mathcal{X}} g(x) \leq M.$ *Let* $M_\delta = c^2 \log(\frac{c}{\delta})$, *where* $c$ *is the same constant in Lemma 4. Then,* $\|\hat{g} - g_0\|_{L_4(\mathcal{D}_s^x)}^4 \leq (M + M_\delta)^2 \|\hat{g} - g_0\|_{L_2(\mathcal{D}_s^x)}^2.$

*Proof of Lemma 5.*

$$
\begin{aligned}
\|\hat{g} - g_0\|_{L_4(\mathcal{D}_s^x)}^4 &= \int_{\mathcal{X}} (\hat{g}(x) - g_0(x))^4 p_s(x) dx = (M + M_\delta)^4 \int_{\mathcal{X}} \left(\frac{\hat{g}(x) - g_0(x)}{M + M_\delta}\right)^4 p_s(x) dx \\
&\leq (M + M_\delta)^4 \int_{\mathcal{X}} \left(\frac{\hat{g}(x) - g_0(x)}{M + M_\delta}\right)^2 p_s(x) dx = (M + M_\delta)^2 \|\hat{g} - g_0\|_{L_2(\mathcal{D}_s^x)}^2
\end{aligned} \tag{101}
$$

$\qquad \square$

# H    Proof of Theorem 5 in the Main Manuscript

**Theorem 5.** *Suppose* $\forall g \in \mathcal{G}, \sup_{x \in \mathcal{X}} g(x) \leq M.$ *Then,* $\|\hat{g} - g_0\|_{L_4(\mathcal{D}_s^x)} \leq \mathcal{O}\left(c^2 \log(\frac{c}{\delta}) \lambda_{n_{s1}, n_t}^{\frac{1}{4}}\right)$ *with probability at least* $1 - \delta$, *where* $c$ *is a constant.*

*Proof of Theorem 5.* By Lemma 5, we have

$$
\begin{aligned}
\|\hat{g} - g_0\|_{L_4(\mathcal{D}_s^x)} &\leq \left(M + c^2 \log(\frac{c}{\delta})\right)^{\frac{1}{2}} \|\hat{g} - g_0\|_{L_2(\mathcal{D}_s^x)}^{\frac{1}{2}} \\
&= \mathcal{O}\left(\left(M + c^2 \log(\frac{c}{\delta})\right)^{\frac{1}{2}} \cdot \left(c^2 \log(\frac{c}{\delta})\right)^{\frac{1}{2}} \lambda_{n_{s1}, n_t}^{\frac{1}{4}}\right) \text{(Lemma 4)} = \mathcal{O}\left(c^2 \log(\frac{c}{\delta}) \cdot \lambda_{n_s, n_t}^{\frac{1}{4}}\right)
\end{aligned} \tag{102}
$$

$\qquad \square$

# I    Related Work

## I.1    Covariate Shift and Sample Selection Bias

The situation where training and test distributions are different is called *dataset shift* (Moreno-Torres et al., 2012), in which generalization guarantees on the test distribution are impossible without further assumptions (Ben-David et al., 2010). Well-chosen assumptions, which can be categorized into three classes: *covariate*, *prior*, and *concept* shift (Moreno-Torres et al., 2012; Amos, 2009), can make possible algorithms with non-vacuous performance guarantees.

There are several possible causes for covariate shift, out of which this section mentions the two we deem most important: the *missing at random (MAR)* case in the sample selection bias and non-stationary environments. Sample selection bias (Cortes et al., 2008) relies on a model of the data generation process. Test instances are drawn from $p_t(x)$. Training instances are drawn by first sampling $x$ from the test distribution $p_t(x)$. A selector variable $s$ then decides whether $x$ is moved into the training set ($s = 1$) or moved into the rejected set ($s = 0$). For instances in the training set, a label is drawn from $p(y|x)$. For the instances in the rejected set, the labels are unknown. A typical scenario for sample selection bias is credit scoring. The labeled training sample consists of customers who were given a loan in the past and the rejected sample are customers that asked for but were not given a loan. New customers asking for a loan reflect the test distribution. The true but unobserved test distribution $\mathcal{D}_t$ can be expressed by

$$p_t(x, y) = \frac{p(x, y|s = 1)p(s = 1)}{p(s = 1|x, y)} = \frac{p(s = 1)}{p(s = 1|x, y)} p_s(x, y) \tag{103}$$

In the MAR case, the selector variable is only dependent on $x$, but not on $y$; that is, $p(s = 1|y, x) = p(s = 1|x)$. Therefore, the conditional distribution of the labels given features is the same for both the training and the test data:

$$p_s(y|x) = p(y|x, s = 1) = \frac{p(x)p(y|x)p(s = 1|x, y)}{p(x)p(s = 1|x)} = p_t(y|x)$$

The MAR bias leads to covariate shift. The second cause for covariate shift appears when the training environment is different from the test one, whether it is due to a temporal or a spatial change (Jang et al., 2022; Jr. et al., 2014; Vovk, 2020; Podkopaev & Ramdas, 2022).

## I.2    Density Ratio Estimation

Empirical risk minimization may fail under covariate shift due to the difference between the training and test distributions. Importance (often referred to the ratio of test and training instance densities) weighting is used to mitigate the influence of covariate shift (Shimodaira, 2000; Sugiyama & Müller, 2005; Sugiyama et al., 2007a; Zadrozny, 2004), which leads to IWERM. Therefore, how to estimate the density ratio accurately becomes the key to success of covariate shift adaptation.

There are two types of density ratio estimation algorithms. The first one is to first separately estimate the two probability densities, and then take the ratio of the estimated densities. However, density estimation is known to a hard problem (Vapnik, 1998; Wolfgang et al., 2004; Kanamori et al., 2010). Therefore, this naive approach may not be effective. The second one is to directly estimating the density ratio without estimating the densities. Various direct density-ratio estimation methods can be divided into four types: Moment Matching, Probabilistic Classification, Density Fitting and Density-Ratio Fitting (Masashi et al., 2012). Of all the direct methods, KLIEP (Sugiyama et al., 2008) and KuLSIF (Kanamori et al., 2012) are two commonly used ones which are computationally efficient and comparable to others in terms of performance (Yamada et al., 2011; Menon & Ong, 2016; Liu et al., 2017; Kumagai et al., 2021; Kato & Teshima, 2021; Tiao et al., 2021; Zhang et al., 2023). We include them as our analysis.

## I.3    The Analysis of Effect of Density Ratio Estimation Error

The closest previous work to ours is Cortes et al. (2008). They consider the problem of determining to what extent the error in the weight estimation can affect the accuracy of the hypothesis returned by a

weight-sensitive learning algorithm in MAR bias problem. Their theoretical results differ from our work in three aspects.

First, there is a key difference between MAR bias and covariate shift. The MAR bias imposes a stricter criterion on the relationship between the training distribution and the test distribution than covariate shift. That is to say, there are some instances of covariate shift that cannot arise from MAR bias, but every instance of MAR bias can be modeled as covariate shift. Consider the problem of data analysis using a brain-computer interface, where the distribution over incoming signals is known to change as experiments go on (subjects tire, the sensor setup changes, etc.). In this example, a covariate shift that cannot be represented by sample selection bias occurs.

We further illustrate the difference between MAR bias and covariate shift in terms of the correction method. The MAR bias correction technique commonly used in machine learning consists of reweighting the loss on training examples to closely reflect the test distribution. Proposition 2 illustrates that minimizing the loss on examples weighted by $p(s = 1|x)^{-1}$ in fact minimizes the expected loss with respect to $\mathcal{D}_t$.

**Proposition 2** ((Cortes et al., 2008; Zadrozny, 2004)). *The expected loss with respect to $\mathcal{D}_t$ is proportional to the expected loss with respect to $\mathcal{D}_s$ with weights $p(s = 1|x)^{-1}$ for the loss incurred by each $x$, provided that the support of $\mathcal{D}_t^x$ is contained in the support of $\mathcal{D}_s^x$:*

$$\mathbb{E}_{z \sim \mathcal{D}_t}[\ell(h(x), y)] \propto \mathbb{E}_{z \sim \mathcal{D}_s}[\frac{1}{p(s = 1|x)}\ell(h(x), y)] \tag{104}$$

When the model is implemented, correcting for MAR bias amounts to estimating $p(s = 1|x)$. This estimation can be accomplished by building a classifier that discriminates the training against the rejected examples. The target model is then learned by following Proposition 2 and weighting training examples by $p(s = 1|x)^{-1}$. No test examples drawn directly from $D_t^x$ are needed to train the model; only labeled selected and unlabeled rejected examples are required. This is in contrast to the covariate shift model that requires examples drawn from the test distribution, but no selection process is assumed and no rejected examples are needed. Covariate shift models can be applied to learning under MAR bias by treating the selected examples as a labeled sample and the union of selected (ignoring the labels) and rejected examples as an unlabeled sample. In contrast, MAR bias models cannot be applied to learning under covariate shift problems that are not caused by MAR bias. Compared with Cortes et al. (2008), we consider a more general case in which covariate shift is analyzed no matter what the cause is.

Second, the risks analyzed in Cortes et al. (2008) and this paper are different. Let $\hat{h}_{g_0}$ denote the predictor returned by IWERM$(\mathcal{H}, S_{s2}; g_0)$. Theorems 2 and 4 in Cortes et al. (2008) provide bounds on the difference between the generalization error of $\hat{h}_{g_0}$ and $\hat{h}$ (i.e., $\tilde{L}_{\mathcal{D}_s}(\hat{h}, g_0) - \tilde{L}_{\mathcal{D}_s}(\hat{h}_{g_0}, g_0)$). In contrast, we consider the difference between the generalization error of the hypothesis returned by IWERM based on the estimated density ratio and the best one in the hypothesis class (i.e. $\tilde{L}_{\mathcal{D}_s}(\hat{h}, g_0) - \tilde{L}_{\mathcal{D}_s}(h^\star, g_0)$). For the weight estimation error, they only consider the weights for the training sample, while our analysis consider the difference between the real density ratio function and the estimated density ratio function. Therefore, the analysis in Cortes et al. (2008) is orthogonal to our work.

Third, the techniques of deriving bounds are different. A *weighted sample $S_{\mathcal{W}}$* is a training sample $S_s$ that is augmented with a non-negative weight $\mathcal{W}_i$ for each example $z_i$. The sample weight $\mathcal{W}$ of $S_{\mathcal{W}}$ defines a distribution over $S_s$. For a fixed learning algorithm $L$, denote by $h_{\mathcal{W}}$ the predictor returned by $L$ for the weighted sample $S_{\mathcal{W}}$. Denote by $d(\mathcal{W}, \mathcal{W}')$ a divergence measure for two distribuutions $\mathcal{W}$ and $\mathcal{W}'$. The key concept for deriving bounds in Cortes et al. (2008) is *distributional $\beta$-stability*.

**Definition 8** (Distributional $\beta$-Stability (Cortes et al., 2008)). *A learning algorithm $L$ is said to be distributional $\beta$-stable for the divergence measure $d$ if for any two weighted samples $S_{\mathcal{W}}$ and $S_{\mathcal{W}'}$,*

$$\forall z \in \mathcal{Z}, |\ell(h_{\mathcal{W}}(x), y) - \ell(h_{\mathcal{W}'}(x), y)| \leq \beta d(\mathcal{W}, \mathcal{W}') \tag{105}$$

Thus, an algorithm is distributionally stable when small changes to a weighted sample's distribution, as measured by a divergence $d$, result in a small change in the loss of the predictor at any example. Proposition 2 in Cortes et al. (2008) shows that kernel-based regularization algorithms are distributionally $\beta$-stable.

**Proposition 3** (Proposition 2 in Cortes et al. (2008))**.** *Let $L$ be a distributionally $\beta$-stable algorithm and let $h_{\mathcal{W}}(h_{\mathcal{W}'})$ denote the hypothesis returned by $L$ when trained on the weighted sample $S_{\mathcal{W}}$ (respectively $S_{\mathcal{W}'}$). Then the following holds*

$$|L_{\mathcal{D}_t}(h_{\mathcal{W}}) - L_{\mathcal{D}_t}(h_{\mathcal{W}'})| \leq \beta d(\mathcal{W}, \mathcal{W}'). \tag{106}$$

In contrast, we demonstrate that if the true risk satisfies a condition called *derivative vanishing* property, the impact of the density ratio estimation error on the excess risk bound of the two-step covariate shift adaptation algorithm is of the fourth order.

Another work relevant to ours is Reddi et al. (2015). To address the issue that plagues many covariate shift correction algorithms, namely that the variance increases considerably whenever samples are reweighted, this paper proposes a doubly robust estimator using the unweighted solution as a variance-reducing proxy for the correct weighed solution. Moreover, they analyze the generalization bounds for the Penalized Risk Minimization (PRM) estimation algorithm based on a meta theorem. However, they only give the generalization bound of the final hypothesis. The effect of the density ratio estimation error on the learning algorithm is not explicitly illustrated by Reddi et al. (2015).

