# OpenReview forum: "Quantitative Analysis of the Effect of Density Ratio Estimation in Covariate Shift Adaptation"
_TMLR — Accepted by TMLR_

### Review · Reviewer_2KMo · 2025-11-09

**Summary Of Contributions:**

The paper aims to clarify the effect of the error in the density ratio estimation on the training of a model for covariate shifts.
In machine learning problems, the distributions between training and test samples may be different, and the covariate shift is an assumption in the difference; the covariate shift assumes that the distribution of the input variables between the training and test samples are different ($p\_s(x) \\neq p\_t(x)$), but the distributions of the output variables conditioned by the input variables are unchanged ($p\_s(y|x) = p\_t(y|x)$). In such a case, it is known that we have only to train the model by, for each training sample, we set the weight on the loss for the sample by the density ratio of the test distribution to the training distribution at the sample.
However, since the true density ratio is usually unknown and we therefore need to estimate it, the error of the estimation affects the performance for the test dataset.
The paper showed the following facts. Firstly, they showed that an upper bound of the difference in the loss (defined by the loss function) between case using the true density ratio and the estimated density ratio is quartic of the possible difference caused by the density ratio estimation algorithm (not the difference of the result of the estimation algorithm; see Definition 1 and Corollary 1). Then, for two density ratio estimation algorithms KLIEP and KuLSIF, they showed that the upper bound is reduced in $\\mathcal{O}(n^{a})$ ($-1 < a < 0$), which was worse than a desirable order $\\mathcal{O}(n^{-1})$.

**Audience:**

Yes

**Audience Explanation:**

Covariate shift is extensively studied as an assumption of the distribution changes between training and test distributions, and novel analyses have been made. Also, they analyzed two existing density ratio estimation methods to derive the rate of the error reduction with respect to training sample size. They are fundamental things to be analyzed.

**Broader Impact Concerns:**

The reviewer did not find any specific concerns.

**Claims And Evidence:**

No

**Claims Explanation:**

Analyses made in the current paper themselves look sound, but there were several points that are not sufficiently convincing to the reviewer due to the lack of desired analyses. Also, the experiment was not sufficient to convince the reviewer. Please see "Requested changes" part for details.

**Requested Changes:**

## Points critical to securing my recommendation for acceptance

- Section 2.3, Definitions 1: It defines $\\|g\\|\_\\mathcal{G} = \\|g\\|\_{L\_4(\\mathcal{D}\_s^x)}$, but in the preceding sentence $\\mathcal{G}$ is defined as a density ratio function class. What does the definition intend? Does it just intend that the norm of $g\\in\\mathcal{G}$ is calculated by $\\|g\\|\_{L\_4(\\mathcal{D}\_s^x)}$?
- Section 3, Definition 3: Since the definition is given for $\\partial\_t T(t)[t^\prime]$, the definition of the expression $\\partial\_a T(b)[c]$ when $a\neq b$, or the expression $\\partial\_a T(b)[c]$ when $c$ includes the variable $a$ are not clear (but used in many places like Assumption 1). Please provide clear definitions even for such cases.
- Section 4.1, Theorem 3 (including conclusions of Sections 4.2 and 4.3): can the $\\Delta$ that attains "inf" be analytically or computationally identified? If difficult, can its lower (upper?) bound be computed?
- Section 4.2, end of the section (same for Section 4.3): It states that "It implies that there is still a distance between the prediction error rate that can be achieved by KLIEP and fast rate, i.e., $\\mathcal{O}(n^{-1})$", but is the rate $\\mathcal{O}(n^{-1})$ is considered as sufficiently fast in this field? If so, please add brief explanation about this fact and/or add references about existing discussions on the rate.
  - Related to the above, are KLIEP and KuLSIF almost the best existing methods of density ratio estimations under covariate shift in terms of the convergence rate?
- Section 5: Experiment with only one model and one dataset is too limited to support the proposed method. Consider adding for both, and (if the page is limited) show one result in the main paper and others in the appendix.
- Section 5: It states that "It is illustrated that the test error of the predictor returned by EIWERM for most of the $\\lambda$ values decreases when NMSE decreases", but the reviewer is not convinced with this consideration.
  - The relationship that smaller NMSE of density ratios yields smaller test error can be seen when $\\lambda$ is relatively small (near to 0). However, to make the sample weights in the training being similar to true density ratio, larger $\\lambda$ (near to 1) is needed. This discrepancy made the reviewer unconvinced. Are there any plausible reasons why such results are produced?

## Points that simply strengthen the work in my view

- Overall: Please add references for functional derivatives (e.g., in Definition 3) and their theorems (e.g., Proposition 1 in Appendix B). It seems that they are not so obvious for researchers in this field.
- Section 2.1: It looks that the term "starhull" is not so standard. (The reviewer heard the term for the first time, and there was no result in Google search for the term.) It seems better to "define" the term, for example, "We define the *starhull* of $x \\in \\mathcal{X}$ as $\\mathrm{star}(\\mathcal{X}, x) = \\{ t\\cdot x + (1 - t)\\cdot x^\\prime \\mid x^\\prime \\in \\mathcal{X}, t \\in [0, 1]\\}$."
- Section 2.3, Definitions 1 and 2: These definitions are the *assumption of the existence* of a DRE or an IWERM algorithm satisfying the inequality. So, it is better to write like "Given $S\_{s\_1}\\sim\\mathcal{D}\_s, S\_t\\sim\\mathcal{D}\_x$, let the algorithm $DRE(\\mathcal{G}, S\_{s\_1}, S\_t)$ can output a density ratio function $\\hat{g}$,".
- Section 3, Assumption 3: In the first expression $(\\mathbb{E}[(2 g\_0(x) h(x))^2])^{\\frac{1}{2}}$, for what distribution the expectation $\\mathbb{E}$ is taken? Please specify. (Perhaps $\\mathbb{E}_{z\\sim\\mathcal{D}_s}$?)
- Section 3, Remark: It states that "This suggests that if the desired oracle rate", but what rate does it intend? (Perhaps the rate of hypothesis's excess risk?)
- Section 3.1, Lemma 1: When we use the notation $\\nabla\_\\zeta$ or the like, if the variable name is replaced, it becomes difficult to identify the variable with which the gradient is taken. For example of Point 1 of Lemma 1, instead of writing $\\nabla\_\\zeta \\nabla\_\\gamma \\tilde{\\ell}(h^\*(x), g\_0(x); z)$, it is clearer to write as $\\nabla\_\\zeta \\nabla\_\\gamma \\tilde{\\ell}(\\zeta, \\gamma; z)|\_{\\zeta = h^\*(x), \\gamma = g\_0(x)}$.
- Section 4.1, Theorem 3 (including conclusions of Sections 4.2 and 4.3): As far as the reviewer's understanding, $N_X(\\Delta, \\mathcal{F})$ is a decreasing function with respect to $\\Delta$. So, instead of defining "$\\inf\_{\\Delta\\in[0, D],\~N_X(\\Delta,\\mathcal{F})\\geq 10}(\mathrm{omitted})$ where $D = \\sup\_{w, w^\\prime \\in \\mathbb{B}(0, 1)} \\sup\_{x\\in\\mathcal{X}} |\\mathrm{sig}(\\langle w, x \\rangle) - \\mathrm{sig}(\\langle w^\\prime, x \\rangle)|$", defining "$\\inf\_{\\Delta\\in[0, D]}(\mathrm{omitted})$ where $D = \\min\\{ \\mathrm{arg}\\max_{\\Delta > 0} N_X(\\Delta,\\mathcal{F})\\geq 10,\~ \\sup\_{w, w^\\prime \\in \\mathbb{B}(0, 1)} \\sup\_{x\\in\\mathcal{X}} |\\mathrm{sig}(\\langle w, x \\rangle) - \\mathrm{sig}(\\langle w^\\prime, x \\rangle)| \\}$" looks more straightforward.
- Section 5: It states that "We use values $\\sigma\\in\\{0.1, 0.2, 0.8\\}$ to vary the quality of the density ratio estimates produced by KLIEP", but what is $\\sigma$? It looks that it is not stated anywhere in the paper; please specify the definition.
- Appendix B, just before expression (54): It states that "$\\beta\_1/\\lambda\\geq 1$ without loss of generality", but it implies that we need to choose $\\beta\_1$ and $\\lambda$ satisfying $\\beta\_1/\\lambda\\geq 1$ *before applying Theorem 1*. A possible workaround is, in Theorem 1, to replace all factors $\\frac{\\beta\_1}{\\lambda}$ in expression (9) (and also (10)) with $\\max\\{ \\frac{\\beta\_1}{\\lambda}, 1 \\}$.

---

> ### Author Response · Authors · 2025-11-23
> **Response to Reviewer 2KMo-- Part 1**
>
> Dear reviewer 2KMo, Thank you for your helpful feedback and insightful questions. Please find our detailed answers below:
>
> **Q1**. Section 2.3, Definitions 1: It defines $||g|| _{\mathcal{G}}=||g|| _{L_4(\mathcal{D}_s^x)}$,  but in the preceding sentence $\mathcal{G}$ is defined as a density ratio function class. What does the definition intend? Does it just intend that the norm of $g\in \mathcal{G}$ is calculated by $||g|| _{L_4(\mathcal{D_s^x})}$ ？
>
> **A1**. Thanks for your comments. In Definition 1 of Section 2.3, we assume the function spaces $\mathcal{G}$ are equipped with norm $||\cdot|| _\mathcal{G}$. The norm takes the functional $L_4$-norm $\||g\|| _{L_4(\mathcal{D_s^x})}$, which is defined in Section 2.1. In other words, it just intend that the norm of $g\in \mathcal{G}$ is defined by $||g|| _{L_4(\mathcal{D_s^x})}$.
>
> **Q2**. Section 3, Definition 3: Since the definition is given for $\partial_{t}T\left(t\right)[t^\prime]$, the definition of the expression $\partial_{a}T\left(b\right)[c]$ when $a\ne b$ , or the expression $\partial_{a}T\left(b\right)[c]$ when $c$ includes the variable $a$ are not clear (but used in many places like Assumption 1). Please provide clear definitions even for such cases.
>
> **A2**. In Definition 3, the directional derivative of the functional $T(t)$ in the direction $t^\prime$, denoted by $\partial_{t}T\left(t\right)[t^\prime]$, is defined to be,
>
> $$
> \partial _{t}T\left(t\right)[t^\prime] = \frac{d}{d\mu }T\left(t+\mu t^\prime\right)\| _{\mu=0}\\\
> = \lim\limits _{\delta\to 0} \frac{T(t+(0+\delta)t^\prime) - T(t+0t^\prime)}{\delta}
> =\lim\limits _{\delta\to 0} \frac{T(t+\delta t^\prime) - T(t)}{\delta}
> $$
>
>   ,which measures the rate at which $T(t)$ changes with respect to $t$ in the direction $t^\prime$.
>
> We also mention the partial derivatives when considering a functional in two arguments in Definition 3. Let $Q = T(h,g)$.
>
>
> 1.The partial derivative of $T$ with respect to $h$ in the direction $t_1^\prime$ is
> $$
> \partial _hT(h,g)[t_1^\prime] = \frac{d}{d\mu }T\left(h+\mu t_1^\prime,g\right)\Big| _{\mu=0}
>  \\ =\lim\limits _{\delta\to 0} \frac{T(h+\delta t_1^\prime,g) - T(h,g)}{\delta}
> $$
> 2.The partial derivative of $T$ with respect to $g$ in the direction $t_2^\prime$ is
> $$
> \partial _gT(h,g)[t_2^\prime] = \frac{d}{d\mu }T\left(h,g+\mu t_2^\prime\right)\Big| _{\mu=0} \\
>   =\lim\limits _{\delta\to 0} \frac{T(h,g+\delta t_2^\prime) - T(h,g)}{\delta}
> $$
> 3.The second partial derivative of $T$ with respect to $h$ in the direction $t_1^\prime$ then $h$ in the direction $t_2^\prime$ is $\partial_h(\partial_hT(h,g)[t_1^\prime])[t_2^\prime]$, denoted as $\partial _h^2T(h,g)[t_1^\prime,t_2^\prime]$.
>
> 4.The second partial derivative of $T$ with respect to $h$ in the direction $t_1^\prime$ then $g$ in the direction $t_2^\prime$ is
> $\partial _g(\partial _hT(h,g)[t_1^\prime])[t_2^\prime]$, denoted as $\partial_g\partial_hT(h,g)[t_1^\prime,t_2^\prime]$.
>
> Similar definitions hold for $\partial_g^2T(h,g)[t_1^\prime,t_2^\prime]$ and $\partial_h\partial_gT(h,g)[t_1^\prime,t_2^\prime]$.
>
> The notations in Assumption 1 are somewhat misleading. In Assumption 1, the definition of *Derivative Vanishing Loss* is provided:
>
> >  Assumption 1: $\tilde{L}_{\mathcal{D}_s}\left(h,g\right)$ is derivative vanishing with respect to $\mathcal{H}$ and $\mathcal{G}$, that is,
>
> >$$
> >\begin{equation}
> >\partial_{g}\partial_{h}\tilde{L}_{\mathcal{D}_s}\left(h^{\star},g_0\right)[h-h^\star,g-g_0] = 0\quad\forall{h}\in \mathcal{H},\forall{g}\in\mathcal{G}.
> >\end{equation}
> >$$
>
> $\partial_{g}\partial_{h}\tilde{L}_{\mathcal{D}_s}\left(h^{\star},g_0\right)[h-h^{\star},g-g_0]$ is the second partial derivative of the risk function with respect to the first argument in the direction $h-h^\star$ at $h^\star$, then the second argument in the direction $g-g_0$ at $g_0$. The $g$ and $h$ in $\partial_g\partial_h$ only implies the argument with respect to which the derivative is taken. However, the $h$ and $g$  in $[h-h^\star, g-g_0]$ is any given $h\in \mathcal{H}$ and $g\in \mathcal{G}$, respectively.
>
> **Q3**. Section 4.1, Theorem 3 (including conclusions of Sections 4.2 and 4.3): can the $\Delta$ that attains "inf" be analytically or computationally identified? If difficult, can its lower (upper?) bound be computed?
>
> **A3.** The $\Delta$ is introduced to bound the Rademacher average associated with $A = \{(h(x_1)-y_1,h(x_2-y_2),\dots,h(x_n)-y_n):h\in \mathcal{H}\}$ as illustrated in Eq.(90) of Appendix E. By approximating the set $\mathcal{F}$ up to some accuracy $\Delta$, we replace the supremum over $\mathcal{F}$ by a finite maximum over the $\Delta$-covering set, plus an approximation error that scales proportionally with $\Delta$, as illustrated in Eq. (92). The first part $2\Delta$ increases as $\Delta$ increase, while the second part  $\frac{2B\sqrt{ d\log\left(1+\frac{B}{2\Delta}\right)}}{\sqrt{n}}$ decreases. Therefore, there exists an intermediate value of $\Delta$ that minimizes their sum.

---

> ### Author Response · Authors · 2025-11-23
> **Response to Reviewer 2KMo-- Part 2**
>
> **Q4.** Section 4.2, end of the section (same for Section 4.3): It states that "It implies that there is still a distance between the prediction error rate that can be achieved by KLIEP and fast rate, i.e., $\mathcal{O}(n^{-1})$", but is the rate $\mathcal{O}(n^{-1})$ is considered as sufficiently fast in this field? If so, please add brief explanation about this fact and/or add references about existing discussions on the rate.
>
> - Related to the above, are KLIEP and KuLSIF almost the best existing methods of density ratio estimations under covariate shift in terms of the convergence rate?
>
> **A4.** The speed with which a learning algorithm converges as it is presented with more data is a central problem in machine learning. Alexey Chervoonenkis and Vladimir Vapnik present quantitiave bounds on the deviation between the empirical and expected risk as a function of the sample size $n$ [1]. According to [1], in Vapnik's book co-authored by Chervonenkis [2], they present 'slow' and 'fast' bounds for Empirical Risk Minimization  (ERM) when used with 0-1 loss. They show that in the realizable or 'optimistic' case (where there is a predictor in the hypothesis class that almost surely predicts correctly, so that the minimum achievable risk is zero) one can achieve fast $O(1/n)$ convergence, as opposed to the 'pessimistic' case where one does not have such a predictor in the hypothesis class and the best uniform bound is $O(1/\sqrt{n})$ (typically so-called *slow* rate, page 127 in [1]). This difference is important because if one is in such a 'fast rate' regime, one can achieve good performance with less data.
>
> Appendix I.2 provides the related works of density ratio estimation. Various direct density-ratio estimation methods can be divided into four types: Moment Matching, Probabilistic Classification, Density Fitting and Density-Ratio Fitting. Of the direct methods, KLIEP [3] and KuLSIF [4] are two commonly used ones which are computationally efficient and comparable to others in terms of performance. Therefore, we include them as our analysis.
>
> [1] Vladimir N. Vapnik. Statistical Learning Theory. John Wiley and Sons, 1998.
>
> [2] Vladimir N. Vapnik and Alexey Ya. Chervonenkis. Theory of Pattern Recognition (in Russian). Nauka, Moscow, 1974. German translation: Theorie der Zeichenerkennung, Akademie Verlag, Berlin, 1979.
>
> [3] Masashi Sugiyama, Taiji Suzuki, Shinichi Nakajima, Hisashi Kashima, Paul von Bünau, and Motoaki Kawanabe. Direct importance estimation for covariate shift adaptation. Annals of the Institute of Statistical Mathematics, 60(4):699–746, 2008.
>
> [4] Takafumi Kanamori, Taiji Suzuki, and Masashi Sugiyama. Statistical analysis of kernel-based least-squares density-ratio estimation. Machine Learning, 86(3):335–367, 2012.
>
>  **Q5.** Section 5: Experiment with only one model and one dataset is too limited to support the proposed method. Consider adding for both, and (if the page is limited) show one result in the main paper and others in the appendix.
>
> **A5.** We acknowledge the reviewer's point. Our theoretical analysis includes both KLIEP and KuLSIF to demonstrate the generality of our framework. Experimentally, we prioritized KLIEP due to the availability of its stable, official codebase, ensuring our results are reproducible and reliable. Implementing KuLSIF robustly is challenging without a standard implementation. The experiments serve as a successful proof-of-concept for our theory.
>
> In the revised manuscript, we will delineate a specific future research agenda: to conduct a comprehensive comparative analysis of DRE estimators (including KuLSIF) within our framework. This is not merely an extension of experiments, but a necessary step to **investigate how the theoretical properties of different estimators translate into empirical performance** in our context, potentially leading to refined guidelines for practitioners.

---

> ### Author Response · Authors · 2025-11-23
> **Response to Reviewer 2KMo-- Part 3**
>
> **Q6.** Section 5: It states that "It is illustrated that the test error of the predictor returned by EIWERM for most of the $\lambda$ values decreases when NMSE decreases", but the reviewer is not convinced with this consideration.
>
> - The relationship that smaller NMSE of density ratios yields smaller test error can be seen when $\lambda$ is relatively small (near to 0). However, to make the sample weights in the training being similar to true density ratio, larger $\lambda$ (near to 1) is needed. This discrepancy made the reviewer unconvinced. Are there any plausible reasons why such results are produced?
>
> **A6.** $0\le \lambda \le 1$ is a hyper-parameter controlling the trade-off between *consistency* and *stability* of EIWERM [1]. It is known that when the training and test densities are different and the true predictor is not included in the model at hand, least-squares learning ($\lambda = 0$ ) is  no longer *consistent* (i.e., the learned predictor does not converge to the optimal one even when the number of training examples goes to infinity). The problem can be overcome by using a least-squares learning weighted by the ratio of test and training densities, i.e., $\min_{\theta} [\frac{1}{n_s}\sum_{i=1}^{n_s}g(x_i^s) \left(\hat{f}(x^s_i;\theta)-y_i^s\right)^2]$ ($\lambda=1$ ). Although the consistency becomes guaranteed by this modification, the weighted least-squares learning tends to have large variance. Therefore, in practical situations with finite examples, a **stabilized** estimator, i.e.,
> $$
> \hat{\theta} _{EIWERM} \equiv \arg \min _{\theta} \left[\frac{1}{n_s}\sum _{i=1}^{n_s}g(x_i^s)^\lambda \left(\hat{f}(x^s_i;\theta)-y_i^s\right)^2\right],
> $$
> would give more accurate estimates.
>
> [1] Masashi Sugiyama and Klaus-Robert Müller. Model selection under covariate shift. In ICANN, 2005.
>
> ## Points that simply strengthen the work in my view
>
> **Q7.** Overall: Please add references for functional derivatives (e.g., in Definition 3) and their theorems (e.g., Proposition 1 in Appendix B). It seems that they are not so obvious for researchers in this field.
>
> **A7.** The directional derivative defined in Definition 3 (called the Gateaux derivative ) is a generalization of the directional derivative in multivariable calculus [1]. Section 1 in [2] provides the definition of the Gateaux derivative.
>
> > Let $\mathcal{B}$ is a Banach space.  The Gateaux derivative of a functional $A(f):\mathcal{B}\to \mathcal{R}$ is defined as
> >
> > $$
> > D_h A(f) = \lim_{\epsilon\to 0}\frac{A(f+\epsilon h)-A(f)}{\epsilon} = \frac{d}{d\epsilon} A (f+\epsilon h)|_{\epsilon = 0}
> > $$
>
> When $\mathcal{B}$ is $\mathcal{R^d}$ this is exactly the same as the directional derivative.
>
> The Gateaux derivative is a one-dimensional calculation along a specified direction $h$. Because it's one-dimensional, we can use ordinary one-dimensional calculus to compute it. Thus, it is usually easy to compute a Gateaux differential even when the space $\mathcal{B}$ is infinite dimensional [3]. The Taylor expansion for the Gateaux directive can be obtained by applying a standard Taylor expansion to the function of the single real variable $T(t) = F(t\cdot g+(1-t)\cdot g^\prime )$, where $t\in [0,1]$, as referenced on page 19 of [4].
>
> [1] Daryoush Behmardi and Encyeh Dehghan Nayeri. Introduction of fréchet and gâteaux derivative. volume 2, pp. 975–980, 2008.
>
> [2] https://my.eng.utah.edu/~cs7640/readings/funtionalDerivatives.pdf
>
> [3] https://www.math.ttu.edu/~klong/5311-spr09/diff.pdf
>
> [4] https://sites.stat.washington.edu/jaw/COURSES/580s/581/LECTNOTES/ch7c.pdf
>
> **Q8.** Section 2.1: It looks that the term "starhull" is not so standard. (The reviewer heard the term for the first time, and there was no result in Google search for the term.) It seems better to "define" the term, for example, "We define the *starhull* of $x\in \mathcal{X}$ as  $\text{star}(\mathcal{X}, x)= \{t\cdot x + (1-t)\cdot x^\prime, t\in [0,1]\}$ ."
>
> **A8.** Thanks for your suggestion. We will revise it in the revised version.
>
> **Q9.** Section 2.3, Definitions 1 and 2: These definitions are the *assumption of the existence* of a DRE or an IWERM algorithm satisfying the inequality. So, it is better to write like "Given $S _{s1}\sim \mathcal{D} _s, S _t\sim \mathcal{D}_x$ , let the algorithm $DRE(\mathcal{G}, S _{s1}, S _t)$ can output a density ratio function $\hat{g}$,".
>
> **A9.** Thanks for the constructive suggestion. We will revise the definitions in the revised version.
>
> **Q10.** Section 3, Assumption 3: In the first expression $(\mathbb{E}[(2g_0(x)h(x))^2])^{\frac{1}{2}}$ , for what distribution the expectation $\mathbb{E}$ is taken? Please specify. (Perhaps$\mathbb{E}_{z\sim \mathcal{D}_s}$?)
>
> **A10.**  Yes. The expectation $\mathbb{E}$ is defined over the distribution $\mathcal{D}_s^x$.

---

> ### Author Response · Authors · 2025-11-23
> **Response to Reviewer 2KMo-- Part 4**
>
> **Q11.** Section 3, Remark: It states that "This suggests that if the desired oracle rate", but what rate does it intend? (Perhaps the rate of hypothesis's excess risk?)
>
> **A11.** Algorithm 1 has a convergence rate, as demonstrated by the upper bounds on the excess risk.
>
> **Q12.** Section 3.1, Lemma 1: When we use the notation $\nabla_{\zeta}$ or the like, if the variable name is replaced, it becomes difficult to identify the variable with which the gradient is taken. For example of Point 1 of Lemma 1, instead of writing $\nabla_\zeta\nabla_\gamma \tilde{\ell}\left(h^\star\left(x\right),g_0\left(x\right);z\right) $, it is clearer to write as $\nabla_\zeta\nabla_\gamma \tilde{\ell}\left(\zeta,\gamma;z\right)|_{\zeta = h^\star\left(x\right),\gamma=g_0(x)}$.
>
> **A12.** Thanks for your suggestion. We will revise it in the revised version.
>
> **Q13.** Section 4.1, Theorem 3 (including conclusions of Sections 4.2 and 4.3): As far as the reviewer's understanding, $N _X(\Delta, \mathcal{F})$ is a decreasing function with respect to $\Delta$. So, instead of defining "inf ${ } _{\Delta \in[0, D], N _X(\Delta, \mathcal{F}) \geq 10}$ (omitted) where $D=\sup _{w, w^{\prime} \in \mathbb{B}(0,1)} \sup _{x \in \mathcal{X}}\left|\operatorname{sig}(\langle w, x\rangle)-\operatorname{sig}\left(\left\langle w^{\prime}, x\right\rangle\right)\right|$ ", defining "inf ${ } _{\Delta \in[0, D]}$ (omitted) where $D=\min \lbrace\arg \max _{\Delta>0} N _X(\Delta, \mathcal{F}) \geq 10, \sup _{w, w^{\prime} \in \mathbb{B}(0,1)} \sup _{x \in \mathcal{X}}\left|\operatorname{sig}(\langle w, x\rangle)-\operatorname{sig}\left(\left\langle w^{\prime}, x\right\rangle\right)\right|\rbrace$ " looks more straightforward.
>
> **A13.** Given that $D$ is a fixed constant while $\Delta$ is a free parameter, we combine all the constraints.
>
> **Q14.** Section 5: It states that "We use values $\sigma \in\{0.1,0.2,0.8\}$ to vary the quality of the density ratio estimates produced by KLIEP", but what is $\sigma$ ? It looks that it is not stated anywhere in the paper; please specify the definition.
>
> **A14.** We apologize for omitting the parameter $\sigma$. The performance of KLIEP depends on the choice of the basis functions $\{\varphi _\ell\left(x\right)\} _{\ell=1}^b$ . Following [1], we use a Gaussian kernel model centered at a subset of $\{x _i^t\} _{i=1}^{n_t}$ for computational efficiency, i.e.,
> $$\hat{g}\left(x\right)=\sum _{\ell=1}^{b} \alpha _{\ell} K _\sigma(x,c _\ell)$$
>  where $K _\sigma(x,x^\prime)$ is the Gaussian kernel with kernel width $\sigma$:
> $$
> K _\sigma(x,x^\prime)=e^{-\frac{\|x-x^\prime\|}{2\sigma^2}},
> $$
>  $c _\ell$ is a template instance randomly chosen from $\{x _i^t\} _{i=1}^{n _t}$ and $b(\le n_t)$  is a prefixed number, as referenced on Eq. (7) of [1].
>
> [1] Masashi Sugiyama, Taiji Suzuki, Shinichi Nakajima, Hisashi Kashima, Paul von Bünau, and Motoaki Kawanabe. Direct importance estimation for covariate shift adaptation. Annals of the Institute of Statistical Mathematics, 60(4):699–746, 2008.
>
> **Q15.** Appendix B , just before expression (54): It states that " $\beta _1 / \lambda \geq 1$ without loss of generality", but it implies that we need to choose $\beta _1$ and $\lambda$ satisfying $\beta _1 / \lambda \geq 1$ before applying Theorem 1. A possible workaround is, in Theorem 1, to replace all factors $\frac{\beta _1}{\lambda}$ in expression (9) (and also (10)) with $\max\lbrace\frac{\beta _1}{\lambda}, 1\rbrace$.
>
> **A15.** Thank you for this constructive comment. You are correct that the current formulation implicitly requires selecting $\beta _1$ and $\lambda$ such that $\beta _1 / \lambda \geq 1$ before applying Theorem 1. A suitable remedy, as you suggested, is to replace all occurrences of $\frac{\beta _1}{\lambda}$ in Equations (9) and (10) with $\max\lbrace\frac{\beta _1}{\lambda}, 1\rbrace$. We will adopt this revision in the updated version and adjust the corresponding proofs accordingly.

---

### Review · Reviewer_q3yo · 2026-01-05

**Summary Of Contributions:**

The paper studies two-step covariate shift adaptation: (i) estimate the density ratio using a density ratio estimation; (ii) run importance-weighted ERM (IWERM) using the estimated ratio. This paper provides a quantitative answer to how does density ratio estimation error propagate to the final learner’s excess risk? The authors formalize the two-step pipeline as a meta-algorithm (Algorithm 1) and show that under a derivative vanishing property of the true risk, the impact of density-ratio estimation error on the excess risk bound is higher order. They further instantiate the framework for KLIEP and KuLSIF, deriving density-ratio error rates and plugging them into the meta bound, plus a toy experiment demonstrating that improved ratio estimation correlates with lower test error.

**Audience:**

Yes

**Audience Explanation:**

1. A clear, well-motivated question with a clean abstraction.
The “meta-algorithm” formalization of two-step covariate shift adaptation makes the analysis modular.

2. The claim that density ratio estimation error can enter the final excess-risk bound, which can be insightful.

3. Concrete instantiations for KLIEP and KuLSIF.
The paper attempts to connect high-level theory to classical DRE methods

**Claims And Evidence:**

No

**Claims Explanation:**

1. The core assumption (“derivative vanishing”) is strong. The main result and derivation of the analysis hinge on Assumption 1. The paper argues square loss satisfies the assumptions under additional conditions, but it is not clear how broadly this property holds for common classification losses and modern models.

2. Scope of the empirical validation is very limited. The experiments section is essentially a toy 2D synthetic classification with KLIEP, varying kernel width and showing test error decreases as NMSE decreases.

3. Practical implications are not fully developed. If the main message is “DRE error enters at higher order,” then the paper should translate this into actionable guidance.

**Requested Changes:**

1. How often does derivative vanishing hold beyond square loss?

2. Can you empirically test the predicted scaling?

3. Add experiments at least one realistic benchmark.

---

> ### Author Response · Authors · 2026-01-13
> **Response to Reviewer q3yo**
>
> Dear Reviewer q3yo, Thank you for your thoughtful review and constructive feedback. Please find our detailed responses below.
>
> **Q1.** The core assumption (“derivative vanishing”) is strong. The main result and derivation of the analysis hinge on Assumption 1. The paper argues square loss satisfies the assumptions under additional conditions, but it is not clear how broadly this property holds for common classification losses and modern models. How often does derivative vanishing hold beyond square loss?
>
> **A1**. We agree that clarifying the scope of this key assumption is critical. Our work provides a complete and rigorous analysis for the square loss. As detailed in Section 3.1 (Lemma 1 & Theorem 2), we prove that under mild boundedness conditions, the square loss naturally and exactly satisfies all assumptions (Assumptions 1–3), including the derivative-vanishing property. The square loss is not only foundational for regression but is also employed in classification via methods like least-squares classifiers, making our results directly applicable to a significant class of problems.
>
> The question of whether this property holds broadly for other common classification losses (e.g., logistic, hinge) or modern, complex model classes is an excellent and important one for future research. It lies beyond the scope of our current paper, which aims to establish a clear theoretical foundation using a canonical and tractable loss. In the revised manuscript, we will add a discussion in Section 7 (Conclusion) to explicitly acknowledge this point and to outline the investigation of the derivative-vanishing condition for other losses as a valuable direction for future theoretical work.
>
> **Q2**. Scope of the empirical validation is very limited. Add experiments at least one realistic benchmark.
>
> **A2**. We acknowledge the reviewer's point. Our theoretical analysis includes both KLIEP and KuLSIF to demonstrate the generality of our framework. Experimentally, we prioritized KLIEP due to the availability of its stable, official codebase, ensuring our results are reproducible and reliable. Implementing KuLSIF robustly is challenging without a standard implementation. The experiments serve as a successful proof-of-concept for our theory.
>
> As a **theoretical paper**, our experiments are deliberately designed on **synthetic data** where the true density ratio $g_0(x)$ is known, allowing us to compute the estimation error (NMSE) and directly validate the relationship between this error and the final test error. This controlled setting is essential for testing our theory. In real-world benchmarks, the true density ratio is unknown, making it impossible to compute the NMSE and directly validate the core quantitative relationship our theory describes.
>
> **Q3**. Practical implications are not fully developed. If the main message is “DRE error enters at higher order,” then the paper should translate this into actionable guidance.
>
> **A3**. Thank you for this suggestion. Our theoretical analysis yields a direct and actionable implication, which is stated in the Remark following Theorem 1 and instantiated in Corollary 1: to achieve an overall excess risk of order $\mathcal{O}(n^{-1})$, the density ratio estimation error $\epsilon_1$ need only converge at the slower rate of $\mathcal{O}(n^{\frac{-1}{2k_2}})$. For the square loss ($k_2 =2 $), this required rate is $\mathcal{O}(n^{-\frac{1}{4}})$.
>
> **This translates into clear, quantitative guidance for practitioners:** the precision required in the density ratio estimation step is less stringent than the final target accuracy for the predictor. A DRE method converging at a rate of roughly $n^{-\frac{1}{4}}$ can be sufficient to support a final predictor converging at the desirable $n^{-1}$ rate. We will make this practical interpretation more prominent in the revised Introduction and Conclusion.
>
> **Q4**. Can you empirically test the predicted scaling?
>
> **A4**. The primary empirical objective of this work is to test the qualitative relationship derived from our theory: that the quality of the density ratio estimate affects the final predictor's performance. The experiments in Section 5 fulfill this objective by showing a consistent trend where improved estimation (lower NMSE) yields lower test error. This successful proof-of-concept provides direct support for the validity of our theoretical analysis in Theorem 1 and Corollary 1.
>
> Thank you again for your insightful comments, which have helped us significantly improve the presentation and impact of our work.

---

### Review · Reviewer_6oqd · 2026-01-07

**Summary Of Contributions:**

The paper provides a theoretical analysis of two-step covariate-shift adaptation methods that combine density-ratio estimation with importance-weighted ERM. Under a derivative-vanishing assumption and technical smoothness/convexity conditions, it shows that the excess risk contribution from density-ratio estimation is fourth order for square loss. The framework is instantiated for KLIEP and KuLSIF using known convergence rates, and a small toy experiment illustrates qualitative trends.

Strengths: clean error decomposition; clear fourth-order result for square loss.
Weaknesses: strong assumptions; limited novelty beyond recombining known rates; very weak empirical validation.

**Audience:**

Yes

**Audience Explanation:**

The results may interest researchers studying covariate shift and importance weighting, but are unlikely to appeal broadly due to limited practical validation.

**Claims And Evidence:**

No

**Claims Explanation:**

The theoretical claims are supported under the stated assumptions, but these assumptions are restrictive and only clearly justified for square loss. Empirical evidence is minimal (single toy example) and does not validate the claimed rates or practical relevance.

**Requested Changes:**

Clarify or narrow the scope of the derivative-vanishing assumption.

Strengthen experiments (non-toy data, include KuLSIF, scaling studies).

Clearly articulate novelty relative to prior work.

---

> ### Author Response · Authors · 2026-01-13
> **Response to Reviewer 6oqd-- Part 1**
>
> Dear Reviewer 6oqd,
>
> Thank you for your thorough review and valuable feedback. Please find our point-by-point responses below.
>
> **Q1**. The theoretical claims are supported under the stated assumptions, but these assumptions are restrictive and only clearly justified for square loss. Clarify or narrow the scope of the derivative-vanishing assumption.
>
> **A1.** Thank you for highlighting this important aspect. We agree that the clarity of the assumption's scope is crucial. The *derivative-vanishing* property (Assumption 1), along with the convexity/smoothness conditions (Assumption 3), are sufficient conditions that enable a clean high-order Taylor expansion. This allows us to isolate and quantify the impact of the density ratio estimation error ($\epsilon_1$) on the final excess risk.
>
> As presented in the manuscript, our analysis in Section 3.1 (Lemma 1 & Theorem 2) immediately follows the main theorem (Theorem 1). There, we prove that these assumptions are naturally and exactly satisfied by the square loss under mild boundedness conditions. Since the square loss is foundational for regression and forms the basis for many practical methods, our theoretical results have direct and broad practical relevance.
>
> In the revised manuscript, we will strengthen the narrative link between Theorem 1 and Section 3.1 to make this connection more immediate and explicit for the reader. We will also refine the wording around Assumption 1 to more clearly present it as a *sufficient condition* that characterizes an important and practical class of problems (with square loss as the canonical example), thereby narrowing and clarifying its scope.
>
> **Q2**. limited novelty beyond recombining known rates; Clearly articulate novelty relative to prior work.
>
> **A2.** We appreciate the opportunity to clarify our novel contributions, which we believe are threefold:
>
> 1. **A Unified Meta-Algorithmic Framework:** We provide the general framework that formulates diverse two-step covariate shift adaptation methods as a single meta-algorithm (Algorithm 1). This framework cleanly decomposes the excess risk into components arising from density ratio estimation and from learning on the weighted sample.
> 2. **Quantifying the Error Propagation:** Under the noted assumptions (satisfied by square loss), Theorem 1 and Corollary 1 establish a precise quantitative relationship: the excess risk scales as $\mathcal{O}(\epsilon_1^{2k_2})$, which is $\mathcal{O}(\epsilon_1^{4})$ for square loss ($k_2 =2 $). A key implication, noted in the Remark following Theorem 1, is that to achieve an overall $\mathcal{O}(n^{-1})$ excess risk, the density ratio estimation error $\epsilon_1$ need only converge at the rate $\mathcal{O}(n^{\frac{-1}{2k_2}})=\mathcal{O}(n^{-\frac{1}{4}})$ (for square loss). This establishes a key theoretical insight from our framework regarding the sufficiency of a slower rate in the first step to guarantee a fast rate in the final excess risk.
> 3. **Explicit Excess Risk Bounds for Classical Estimators:** While convergence rates for KLIEP and KuLSIF are known individually, we  integrate them into this specific framework to derive the resulting excess risk bounds for the final classifier. This analysis explicitly shows that their rates, when used in this two-step pipeline, fall short of the oracle $\mathcal{O}(n^{-1})$ rate, providing a new, comparative perspective on these established methods.
>
> In addition, a distinct theoretical analysis from prior work is systematically detailed in Section 6.2 of our paper. Our analysis differs fundamentally from Cortes et al. 2008 in (i) **problem setting** (general covariate shift vs. sample selection bias), (ii) **risk of interest** (excess risk relative to the optimal hypothesis vs. difference between two empirical hypotheses), and (iii) **analytical technique** (derivative-vanishing property vs. distributional stability). We will revise the Introduction and Conclusion to forefront these points.

---

> ### Author Response · Authors · 2026-01-13
> **Response to Reviewer 6oqd-- Part 2**
>
> **Q3.** Empirical evidence is minimal (single toy example) and does not validate the claimed rates or practical relevance. Strengthen experiments (non-toy data, include KuLSIF, scaling studies).
>
> **A3.** We acknowledge the reviewer's point. Our theoretical analysis includes both KLIEP and KuLSIF to demonstrate the generality of our framework. Experimentally, we prioritized KLIEP due to the availability of its stable, official codebase, ensuring our results are reproducible and reliable. Implementing KuLSIF robustly is challenging without a standard implementation. The experiments serve as a successful proof-of-concept for our theory.
>
> As a **theoretical paper**, our experiments are deliberately designed on **synthetic data** where the true density ratio $g_0(x)$ is known, allowing us to compute the estimation error (NMSE) and directly validate the relationship between this error and the final test error. This controlled setting is essential for testing our theory. In real-world benchmarks, the true density ratio is unknown, making it impossible to compute the NMSE and directly validate the core quantitative relationship our theory describes.
>
> **Q4**. The results may interest researchers studying covariate shift and importance weighting, but are unlikely to appeal broadly due to limited practical validation.
>
> **A4**. Our theoretical analysis yields a direct and actionable implication, which is stated in the Remark following Theorem 1 and instantiated in Corollary 1: to achieve an overall excess risk of order $\mathcal{O}(n^{-1})$, the density ratio estimation error $\epsilon_1$ need only converge at the slower rate of $\mathcal{O}(n^{\frac{-1}{2k_2}})$. For the square loss ($k_2 =2 $), this required rate is $\mathcal{O}(n^{-\frac{1}{4}})$.
>
> **This translates into clear, quantitative guidance for practitioners:** the precision required in the density ratio estimation step is less stringent than the final target accuracy for the predictor. A DRE method converging at a rate of roughly $n^{-\frac{1}{4}}$ can be sufficient to support a final predictor converging at the desirable $n^{-1}$ rate. We will make this practical interpretation more prominent in the revised Introduction and Conclusion.
>
> Thank you again for your insightful comments, which have helped us significantly improve the presentation and impact of our work.

---

### Decision · Action_Editor_aFjL · 2026-02-26

**Recommendation:** Accept as is

**Audience:**

Yes

**Audience Explanation:**

While the paper is somewhat technical in nature, the basic points of inquiry are entirely natural and can be expected to have an audience.

**Claims And Evidence:**

Yes

**Claims Explanation:**

The problem of interest is stated quite clearly; the authors are interested in the impact that a density estimation subroutine has on a subsequent prediction procedure as part of a two-stage learning algorithm. Their results are built around well-established theory, and their main claims related to the theoretical results obtained look to be solid, and the reviewers are in general agreement regarding the theoretical contributions.

Some reviewers were unsatisfied with the empirical analysis and scope of their assumptions, which are valid points to raise that I agree with, but in light of the reviewing guidelines of TMLR (more weight on clarity and solid claims than significance), since the main claims made by the authors are quite precise and theoretical in nature, I believe the evidence provided here is sufficient, and combined with the clarity of the presentation I believe the paper warrants acceptance.